# A dimensionless framework for predicting submarine fan morphology

Abdul Wahab [1] ✉, David C. Hoyal [2], Mrugesh Shringarpure[2] &
Kyle M. Straub [1]

Observations of active turbidity currents at field scale offers a limited scope which challenges the development of theory that links flow dynamics to the morphology of submarine fans. Here we offer a framework for predicting submarine fan morphologies by simplifying critical environmental forcings such as regional slopes and properties of sediments, through densimetric Froude (ratio of inertial to gravitational forces) and Rouse numbers (ratio of settling velocity of sediments to shear velocity) of turbidity currents. We leverage a depth-average process-based numerical model to simulate an array of submarine fans and measure rugosity as a proxy for their morphological complexity. We show a systematic increase in rugosity by either increasing the densimetric Froude number or decreasing the Rouse number of turbidity currents. These trends reflect gradients in the dynamics of channel migration on the fan surface and help discriminate submarine fans that effectively sequester organic carbon rich mud in deep ocean strata.

Submarine fans form on continental slopes and abyssal plains in deepwater settings and are the ultimate sinks in sediment routing systems. They are crafted by turbidity currents, which arguably move more sediment on Earth than any other transport process and their deposits preserve the thickest accumulations of sediment on Earth[1]. Like deltas, the morphologies of submarine fans are quite variable[2–4] and largely result from the dynamics of the channels which move over their surfaces. Unlike deltas, where we have predictive theory for their morphology[5,6], we lack a quantitative and physics-informed theory that successfully predicts the array of observed submarine fan morphologies. This is critical as the surface dynamics that produce these morphologies influence the fidelity of climate change records housed in their strata[7], the frequency and severity of geohazards that threaten subsea infrastructures such as the network of seafloor cables that hosts the majority of global data traffic[8–10], and voluminous geofluid reservoirs[11]. In the last few decades, research has focused on the influence submarine fans have on global climate. Two avenues along this path are the management of marine litter, including the transport of microplastics into deep oceans[12], and burial of particulate organic carbon[13,14].

Turbidity currents are dilute sediment-gravity flows in which the driving gravitational force is supplied by turbulently suspended sediments[15]. These currents can travel for $10^2$ h and up to $10^3$ kilometers in deepwater settings (bathymetry ->4000 m)[16,17], which makes direct monitoring of modern currents challenging. Further complicating the matter, our ability to infer paleo-hydraulic and sediment-transport regimes from the strata of ancient fans is limited. While physical experiments provided insight into interactions of turbidity currents with an antecedent bed topography[18–20] over short timescales, we still lack experimental methods that can produce self-formed channels by fully turbulent flows and which display sufficient mobility to form a fully dynamic fan surface[21,22].

These challenges influence the development of forward numerical models of turbidity currents and their resulting submarine fans. Some of these models carry large computational costs to solve Navier-Stokes equations for complex and fully turbulent turbidity currents. On the other side of the spectrum are rules based reduced complexity models, some of which have been around for decades but are being refined through field measurements and distillation of observations from the high fidelity models[23,24]. While these reduced complexity models

[1]Department of Earth and Environmental Sciences, Tulane University, New Orleans, LA 70118, USA. [2]ExxonMobil Upstream Research Company, Houston, TX 77389, USA. ✉e-mail: awahab@tulane.edu

generate realistic gross scale compensational stacking patterns and lobe-scale geometries over an antecedent surface, they do not account for erosion of the underlying deposits. The lack of spatial and temporal variations in deposition and erosion on the fan surface limit their applicability to unchannelized and purely depositional submarine fans.

We overcome the direct observational, experimental, and modeling limitations that prevent us from linking properties of turbidity currents to the complex morphologies of submarine fans by leveraging a previously developed, parallel process-based numerical model called EMstrata[25,26]. This model is based on the depth-average approximation of flow parameters in 3D Navier-Stokes equations of fully turbulent flows[27–30]. EMstrata solves continuity equations for fluid mass, sediment mass, linear momentum, and turbulent kinetic energy for turbidity currents, in addition to erosion and deposition of cohesive sediments[31] and flocculation of mud particles[32,33] (see methods). The depth-averaged approximations used in this model successfully simulate self-formed channels on fan surfaces that are laterally mobile and avulse through time to generate complex submarine fan morphologies. However, the depth averaged approach means the model does not resolve evolution of surface morphologies tied to the vertical flow structure (e.g., ripples, dunes and anti-dunes). Further, this modeling approach utilizes top hat functions for flow concentration and velocity to determine properties of flows that are stripped by partial confinement, which recent work highlights as limitations when modeling the morphodynamics of submarine fans. Specifically, it has been shown that use of self-similar shape functions that describe flow structure, and which vary as a function of the densimetric Froude Number ($Fr_D$) and Rouse Number ($p$), can aid estimation of mass and momentum fluxes[34]. These limitations noted, EMstrata captures rich dynamics of turbidity currents and their resulting deposits with reasonable computational costs, allowing investigation of hundreds of models.

Here, we posit that submarine fan morphology is largely controlled by the densimetric Froude ($Fr_D = \frac{U}{\sqrt{Rcgh}}$; where $U$ is the down system current velocity, $R$ is the submerged specific gravity of sediment, $c$ is volumetric sediment concentration, $g$ is gravitational acceleration, and $h$ is the current thickness) and Rouse ($p = \frac{w_s}{ku_*}$; where $w_s$ is the sediment fall velocity, $k$ = von Kármán's constant, and $u_*$ is the flow shear velocity) numbers of turbidity currents. More specifically, we hypothesize that flows with low $Fr_D$ and $p$ numbers transport most of their sediment in suspension through a growing network of branching channels and generate distributive fan morphologies. Whereas flows with high $Fr_D$ and high $p$ numbers transport sediment in both suspension and bedload through networks that are often defined by a single channel that avulses over time to generate a radial fan morphology.

We initiate simulations by introducing a depth-averaged turbidity current which traverses an inlet channel into a square domain that is scaled by the advection length of the median grain size in suspension (Fig. 1). The initial surface slope is a dependent parameter defined by a choice of $Fr_D$, clear water entrainment and drag (see methods). The range of $Fr_D$ explored resulted in models with initial slopes between 0.065° and 3.63°, similar to slopes observed on continental margins[35]. Further, the setup of our numerical experiments is designed to explore net depositional submarine fans downstream of canyons or deposits that develop downslope of relatively immobile feeder channels. To isolate the influence of flow hydraulics and sediment-transport capacity on submarine fan shape and dynamics we neglect subsidence and changing climatic forcings (i.e., sea-level, water flux, and sediment supply) within model runs. Simulations are stopped after a defined volume of sediment is released into the model domain.

## Results
### The dimensionless regime
Classification systems exist for submarine fans based on characteristics of the terrestrial feeder system[36], preserved grain sizes[37,38], facies associations[39–41], and preserved morphology[42,43]. Other studies leverage observations that fans on high slopes (e.g., Squamish and Golo fans) are generally small (<10's of km) and sandy[2,14], intermediate slope fans (e.g., Gulf of Cadiz and East Breaks) contain both sand and mud[3,44], and fans that form on low slopes (e.g., Zaire fan) are large (>1000 km) and rich in mud content[4,45]. While the studies that underpin these classification systems report correlations for parameters that influence fan shape, they generally carry tremendous scatter, which limits their predictive power. Many of these classification systems assume links to the hydraulics and sediment-transport capacity of turbidity currents, but we currently lack a predictive framework that explicitly links fan shape to flow processes.

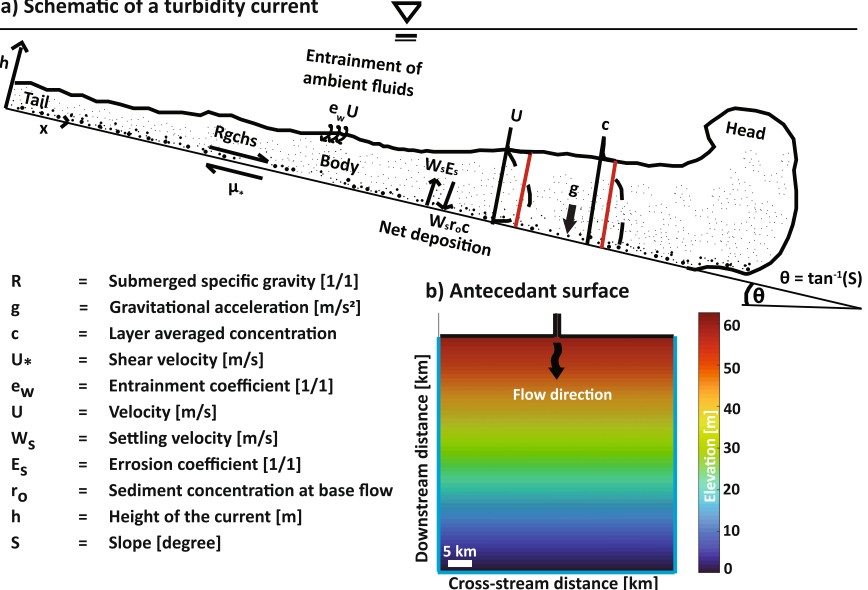

### a) Schematic of a turbidity current

| | | |
|---|---|---|
| R | = | Submerged specific gravity [1/1] |
| g | = | Gravitational acceleration [m/s²] |
| c | = | Layer averaged concentration |
| $U_*$ | = | Shear velocity [m/s] |
| $e_W$ | = | Entrainment coefficient [1/1] |
| U | = | Velocity [m/s] |
| $W_S$ | = | Settling velocity [m/s] |
| $E_S$ | = | Errosion coefficient [1/1] |
| $r_O$ | = | Sediment concentration at base flow |
| h | = | Height of the current [m] |
| S | = | Slope [degree] |

### b) Antecedant surface

**Fig. 1 | Numerical setup. a** Schematic cross-section of a turbidity current with important depth-averaged terms highlighted in red. **b** Example of a model domain [50 × 50 km] with a flat non-erodible slope. Black lines trace upstream boundaries of domain that are closed and reflect flow, whereas blue lines trace boundaries that are open and allow the flow to exit the domain.

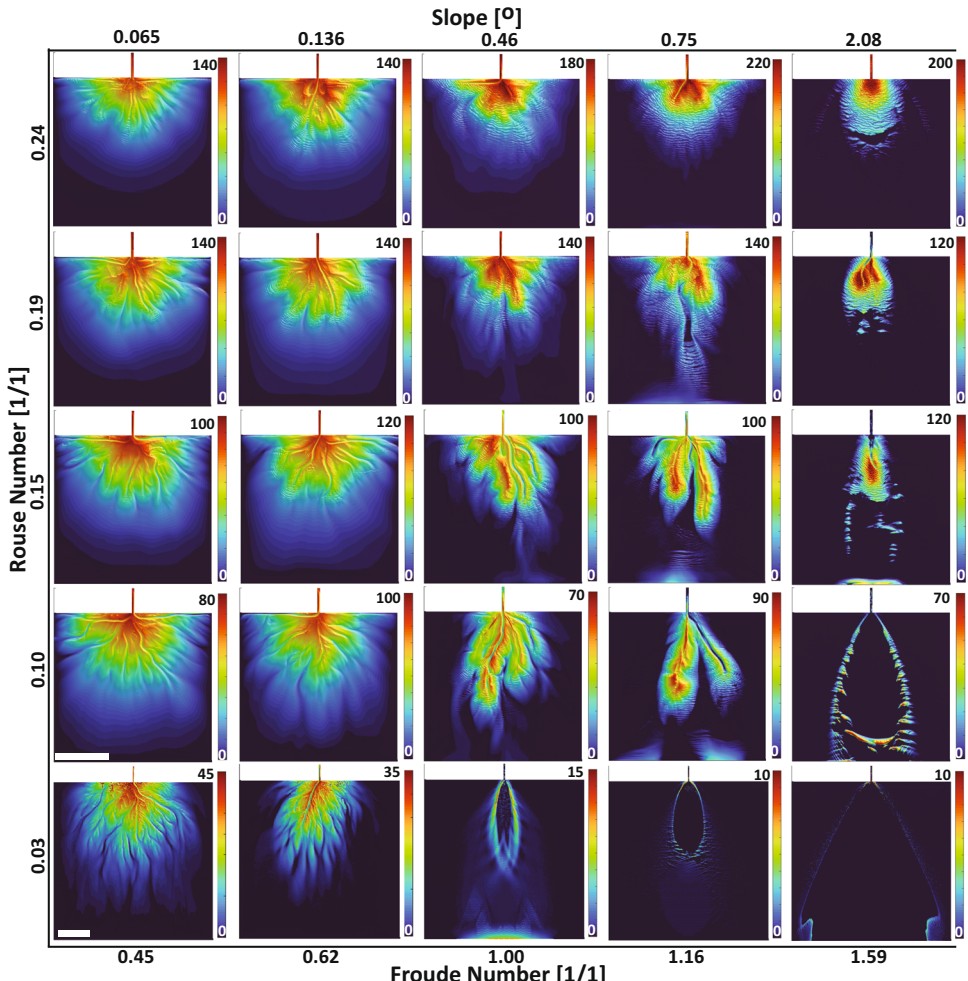

**Fig. 2 | Dimensionless regime diagram.** A subset of simulated fans displays different planform morphologies that result from defined combinations of $Fr_D$ and $p$. Note that range of fan thicknesses displayed in color and the horizontal scale vary between models. White bars denotes 10 km of length and colored bars represents deposit thickness in meters. Models in the last row have domain size of 55 × 50 km whereas the remaining models have a domain size of 35 × 30 km. Additional variables are listed in Supplementary Table 1.

We propose a dimensionless regime where submarine fan dynamics and resulting morphology are defined by two dimensionless variables, $Fr_D$ and $p$ that weigh competing forces in turbidity currents (Fig. 2). The $Fr_D$ quantifies the ratio of inertial to gravitational forces within a flow, while the $p$ describes the ratio of settling velocity of sediments to shear velocity in the flow. To implement this regime, we solve for an antecedent slope, which we set as our initial bathymetric gradient, from a choice of $Fr_D$ as function of bed friction and clear water entrainment[46]. We compute for the median grain diameter in suspension at the inlet from a choice of $p$ as a function of settling velocity. While $Fr_D$ and $p$ dynamically change over the evolving fan surface, we classify fans based on their inlet $Fr_D$ and $p$ values. To capture a wide range of morphodynamic behaviors, we explore ranges in $Fr_D$ and $p$ for a depth-averaged turbidity current (Supplementary Table 1). To avoid either highly erosional or purely depositional flows, we performed sensitivity analyses that guided the range of conditions explored.

**End member morphodynamics**

The simulated submarine fans in our regime diagram have morphologies and surface dynamics that are defined by $Fr_D$ and $p$ and display striking similarity to both modern and preserved examples of submarine fans (Figs. 3–5). In this section, we focus on endmember submarine fans that are crafted by turbidity currents characterized by subcritical ($Fr_D < 1$),

transcritical ($Fr_D \sim 1$) and supercritical ($Fr_D > 1$) flow and defined sediment-transport conditions through $p$. Each numerical submarine fan defined by the aforementioned conditions is then compared to a field counterpart with similar forcing conditions i.e., estimated $Fr_D$ and $p$ (see methods). We note that the absolute scales of some of our simulated fans are small relative to field scale systems, particularly the subcritical end members. This is purely a consequence of the input height of our currents and the model stopping condition, which was picked to limit computational costs for the hundreds of runs we perform. Thus, our goal is not to simulate any particular system, but rather to compare the general morphological complexity of field scale fans with modeled fans that share similar forcings as defined by $Fr_D$ and $p$. To accomplish this, we will also use a dimensionless metric, rugosity, to quantify the morphological complexity of submarine fans.

Our model captures both large and small-scale geomorphic features of subcritical and low $p$ fans that are preserved on low slopes (e.g., Zaire[4,45]) (Fig. 3a–c & Supplementary Movie 1). We observe multiple self-formed sinuous channels bound by prominent levees (Fig. 3b, c). These fans are crafted by channel networks that fluctuate between one-to-many active channels. The channels generate a distributive pattern by avulsing at a variety of distances from the inlet channel (Fig. 3b, c). While we force the simulations with inlet subcritical ($Fr_D < 1$) flow, the flow behavior over the emergent fan is not subcritical everywhere on the fan surface. For example, on the levees,

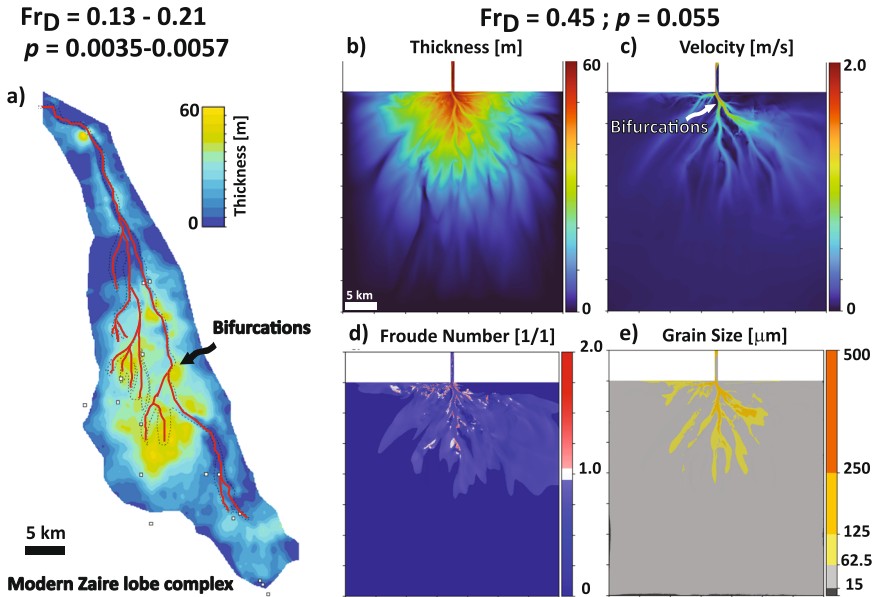

**Fig. 3 | Comparison of a simulated subcritical fan to the modern Zaire fan system, offshore Congo. a** Deposit thickness map of the modern Zaire fan system generated using ZaiAngo surveys[4]. **b** Deposit thickness (m). **c** Velocity field on the active fan surface. **d** $Fr_D$ state on the active fan surface. **e** Grain sizes on the active fan surface. The subcritical fan simulation is characterized by a distributive transport pattern, multiple active and sinuous channels, a hierarchy of bifurcations, and overall mud rich deposit that resembles the Zaire fan[4] (**a**). For the evolution of this modeled subcritical submarine fan see Supplementary Movie 1.

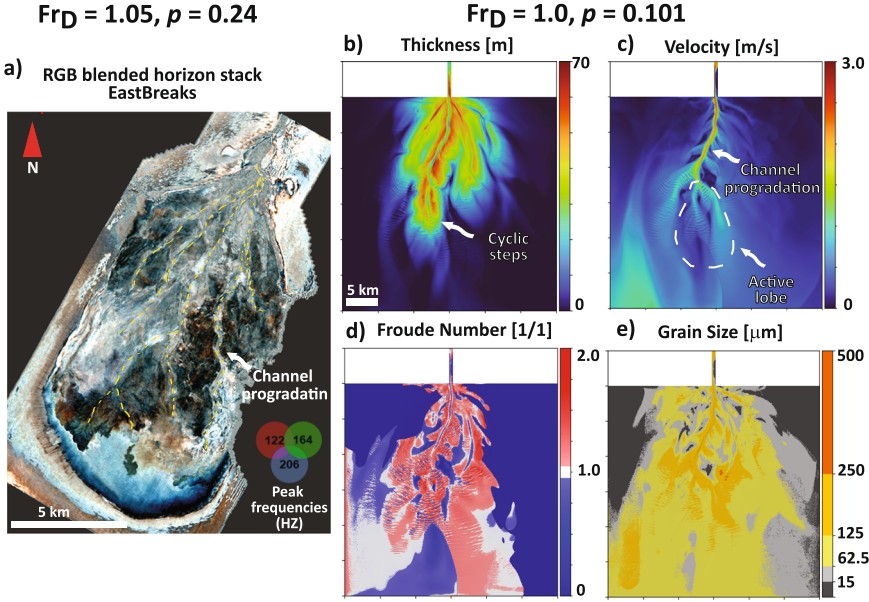

**Fig. 4 | Comparison of a simulated transcritical fan to the Quaternary East Breaks basin-IV submarine fan, offshore Gulf of Mexico. a** Spectrally decomposed seismic horizon of the East Breaks basin-IV submarine fan. The yellow dotted lines highlight channelized bodies that prograded into basin-IV. **b** Deposit thickness (m). **c** Velocity field on the active fan surface. **d** $Fr_D$ state on the active fan surface. **e** Grain sizes on the active fan surface. The transcritical fan simulation (**b–e**) shows a sinuous channel that progrades into the basin (**b–c**), formation of cyclic steps on the active lobe surface (**b**), and distribution of grain sizes on the active fan surface (**e**). For the evolution of this modeled transcritical submarine fan see Supplementary Movie 2.

flow tends to locally change criticality and induce formation of cyclic steps[47] (Fig. 3d). However, the average flow conditions in the inlet channel and the emergent self-formed channels maintain a mean subcritical ($Fr_D < 1$) state. We observe a hierarchy of bifurcations that help partition flux of sediment and water moving through the growing distributive network (Fig. 3b, c). This partitioning is variable and leads to the formation of asymmetric branching channels. In these fans, sand is mainly transported in channels, while silts and muds are deposited on levees, lobes, and distal parts of the outer fan (Fig. 3e).

Transcritical and intermediate $p$ submarine fan surfaces (e.g., Gulf of Cadiz and East Breaks)[3,44] are characterized by self-formed low sinuosity channels that extend in the primary flow direction (Fig. 4a–c & Supplementary Movie 2). The channels are bound by well-formed levees that help maintain the bypass conditions for the majority of their simulated time (Fig. 4d). The channel avulses infrequently after increased aggradation rates within the channel causes a decrease in flow confinement. The newly formed channels extend in the major flow direction, which eventually leads to an ellipsoidal fan morphology. In

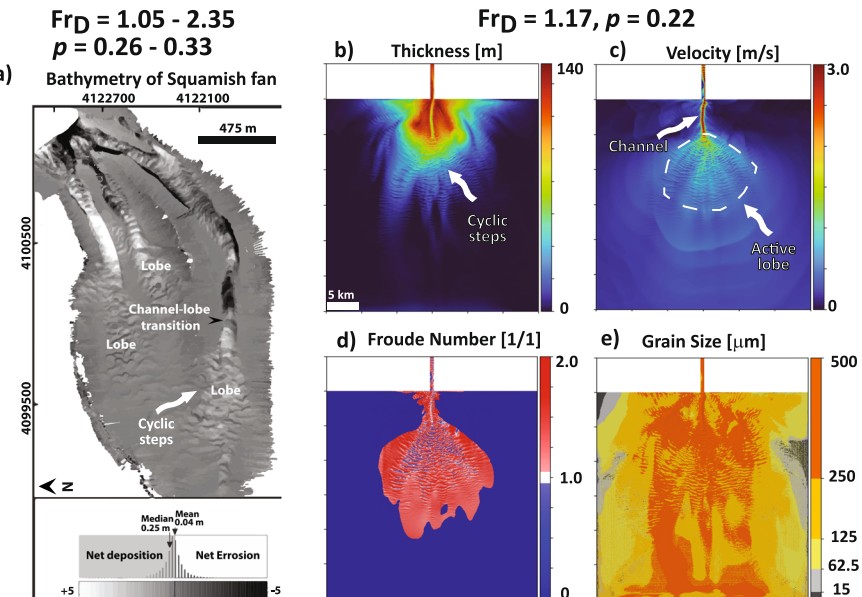

**Fig. 5 | Comparison of a simulated supercritical fan to the modern Squamish fan, offshore British Colombia. a** Bathymetry map of the modern Squamish fan[53]. **b** Deposit thickness (m). **c** Velocity field on the active fan surface. **d** $Fr_D$ state on the active fan surface. **e** Grain sizes on the active fan surface. The supercritical fan simulation (**b–e**) shows characteristic cyclic bedforms (**a**) formed due to hydraulic jumps, a single active and low-sinuosity channel (**c**), and sand rich deposit (**e**) that resembles a high slope modern fan[53] (**a**). For the evolution of this modeled supercritical submarine fan see Supplementary Movie 3.

this type of fan, sand is transported in the channels and deposited on lobes and proximal levees, whereas silt and mud are deposited on the distal parts of the fan surface (Fig. 4e).

Supercritical and high $p$ submarine fans (e.g., Squamish fan)[2] are characterized by a self-formed single low-sinuosity channel (Fig. 5a, b & Supplementary Movie 3). At the channel-lobe transition the flow undergoes a hydraulic jump, where the increased flow height and loss of turbulence lead to deposition of sediments in the form of cyclic steps[48–50] (Fig. 5b–d). These cyclic steps migrate upstream in the channel until the flow confinement decreases and an avulsion is triggered. This type of fan is predominantly sandy because flow shear stresses inhibit deposition of silt and mud (Fig. 5e). The absence of flux partitioning at the channel-lobe transition and avulsions triggered in a single channel system that visits many locations through time induces an overall radial fan shape.

**Submarine fan shape and texture**

The shape of submarine fans and the size of the particles they house provide insight into the hydraulics and sediment-transport capacity of the flows that constructed them. Here we focus on the planform complexity of submarine fans and use regional bathymetric slopes (or slope estimates for paleo-fans) and deposit grain sizes to constrain the $Fr_D$ and $p$ responsible for constructing a given fan. First, we quantify the rugosity of submarine fans as a proxy for the complexity of their morphologies (Fig. 6a). Contrary to deltas, where the shoreline represents an important contour for delta front deposition[6], there is no established contour that reflects the influence of forcing conditions on submarine fan shape. We overcome this by utilizing a deposit thickness contour that delineates the depositional midpoint in submarine fans. Specifically, we use a mass-balance transformation[51], which measures downstream distance in terms of sediment mass lost to deposition in a basin. We define a dimensionless mass extraction parameter, $\chi$, as:

$$\chi(r) = \frac{1}{V_{s,b}} \int_0^L B(r) H_s(r) dr \qquad (1)$$

Where $V_{s,b}$ is the total volume of sediment stored in the basin, $L$ is the total downstream length of a basin, $B$ is the basin width at a radial distance measured from the end of the entrance channel $r$, and $H_s$ is the average thickness of preserved strata along a strike transect defined by $r$. The value of $\chi$ ranges between 0 and 1, for instance, a value of 0.5 implies that half of the sediment stored in a basin is upstream of that location. We compute for a semicircle with an origin at the end of the inlet channel and a radius equivalent to $\chi = 0.5$. We contour the fan based on the mean deposit thickness on this semicircle and evaluate its rugosity as,

$$R(\chi = 0.5) = \frac{1}{N} \sum_{i=1}^{n} \frac{|r_c - r_{mid}|}{r_{mid}} \qquad (2)$$

where $N$ is the number of deposit thickness maps (see Supplementary Note 1), $n$ is number of x–y locations on the semicircle, $r_c$ is the radial distance between the origin of the semicircle and an x–y location on the mean thickness contour, and $r_{mid}$ is the radius of the $\chi = 0.5$ semicircle (Fig. 6a). We exclude models that exhibit near complete sediment bypass.

The simulations show a systematic increase in planform roughness, quantified through rugosity, as one explores from the low $Fr_D$ and high $p$ corner of the regime diagram toward the high $Fr_D$ and low $p$ domain (Fig. 6b). We interpret the joint influence of $Fr_D$ and $p$ on fan rugosity as thus. Given fans of constant $Fr_D$, decreasing $p$ results in longer transport distances for particles in suspension, allowing active and relatively stable channels to quickly prograde. In addition, decreasing $p$ leads to more poorly sorted sediments, due to our inlet suspension criteria (see methods). Like prior research on deltaic channels[6], we propose that an increase in sorting promotes more frequent and asymmetric channel bifurcations, leading to multiple active channels. While this increase in active channels facilitates the delivery of sediment to more fan locations, the stable and asymmetric nature of these channels results in jagged contours with high rugosity values.

Traversing our regime along paths of constant $p$ but increasing $Fr_D$ corresponds to higher shear velocities for flows of equivalent thickness at the inlet, which inhibits deposition of silts and finer sands and promotes the maintenance of fewer active channels. These channels terminate in a single large mouth bar deposit

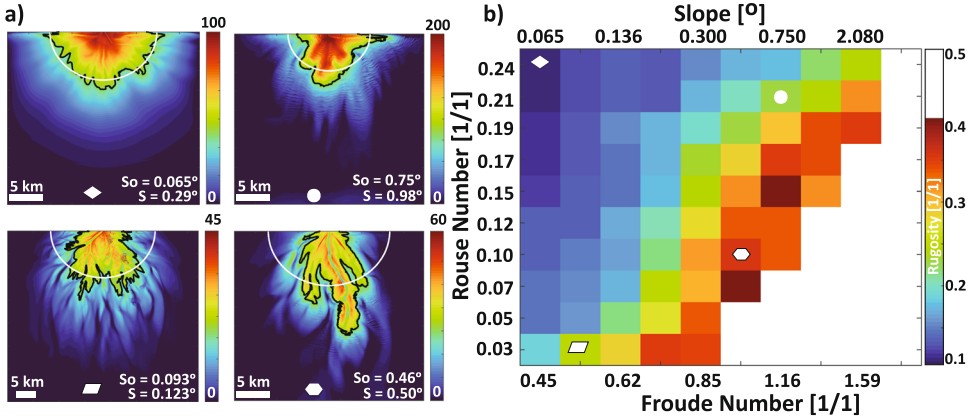

**Fig. 6 | Morphology of simulated fans. a** Deposit thickness maps for select submarine fans show contour used to measure rugosity. The white semicircle defines a distance from the inlet upstream of which half of the sediments in the model domain are stored. The black contour is computed for a mean elevation along the semicircle. Color bars denote thickness [m]. Slopes reported were measured from the entrance channel to the end of the deposit on down system transects. So is the antecedent slope (°) and S(°) is measured at the end of the model run. **b** Matrix shows computed rugosity for all models. White area represents models in which input sediment largely bypassed model domains. Symbols correspond to rugosity measurements for fans displayed in (**a**).

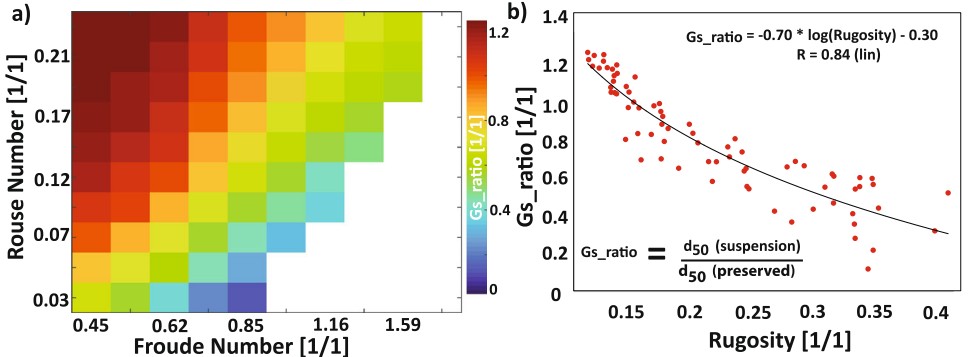

**Fig. 7 | Empirical relationship between ratio of median ($d_{50}$) grain size in transport at the inlet to median grain size preserved in the deposit and fan rugosity. a** Matrix shows grain size ratio (Gs_ratio) computed for all models with a significant deposit. **b** A logarithmic trend is fit to the scatter plot of Gs_ratio and fan rugosity. This trend line allows for estimation of mean grain sizes in suspension for a given $Fr_D$ and $p$ value.

covered with backstepping cyclic steps. This promotes channel mobility through infrequent avulsions caused by the loss of confinement due to the backstepping bedforms. As a result, fan rugosity increases due to the elongation of the relatively active few channels, contrary to the processes responsible for increasing rugosity through decreasing $p$. However, as $Fr_D$ increases past criticality sediment bypass conditions are reached and further increases in $Fr_D$ produce unfavorable conditions for channel initiation and maintenance.

## Surface and subsurface predictions

While rugosity alone is not a unique indicator of $Fr_D$ and $p$, fan rugosity coupled with knowledge of one of these parameters allows for the estimation of the other. For example, the rugosity of modern submarine fans can be estimated with high resolution bathymetric and shallow seismic surveys, whereas for ancient fan deposits it can be estimated with industry-grade 3-D seismic surveys. $Fr_D$ can then be estimated with the regional (paleo)slopes and existing empirical functions for drag coefficients and clear water entrainment[46], as implemented in our model. Fan rugosity and $Fr_D$ can then be used to invert for the $p$ of the inlet flow to a fan. Furthermore, sediment sizes suspended in the inlet flow can be estimated from knowledge of the feeder channel depth. This would involve estimating the shear velocity

of the inlet flow from channel depth and $Fr_D$, followed by use of the $p$ to solve for sediment sizes in transport through a settling velocity.

Alternatively, estimates of flow $p$ in a fan feeder channel coupled with fan rugosity allows for the inversion of $Fr_D$. Here, the challenge is to estimate a paleo-flow $p$ of the feeder channel. This can be done with access to statistics of deposit grain sizes, for example from core measurements. This would entail using an assumption that the largest preserved particles met a Shield's criterion, which could be used to invert for a shear velocity. The $p$ can then be calculated with a characteristic settling velocity of the median particle size in transport. To accomplish this, we found and utilized an empirical relationship (see methods) between fan rugosity and a ratio formed by the sizes of the median particles in transport to the median size in the deposit (Fig. 7a, b).

In the last decade measurements of active flow processes in submarine channels, which were once rare and largely limited to cable breaks, are becoming more common. These data are aiding our characterization of these important transport processes and parameterization of numerical models. However, data collection is still limited to systems that are active today and limited to several highly instrumented settings[16,52,53]. The regime space we propose allows formative flow conditions to be inferred in settings where we still lack active flow measurements, which could aid geohazards risk

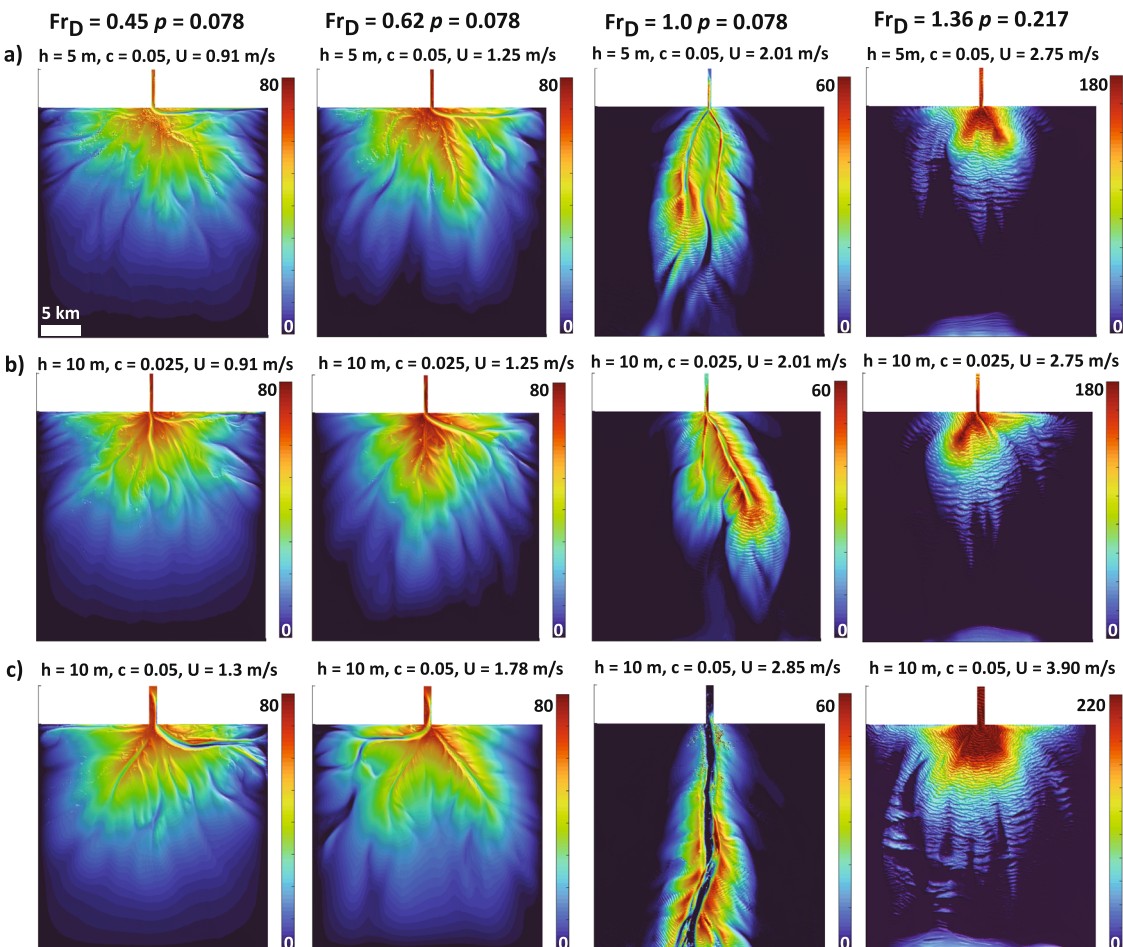

**Fig. 8 | Sensitivity of submarine fan shape to dimensional scales in Froude number ($Fr_D$) of the flow at the inlet. a** Simulated submarine fans in this row display our base case scenario where at the inlet we fix the height of the current (h) and volumetric concentration of suspended sediments (**c**). So by varying Froude number of the flow we only vary the mean velocity (u) at the inlet. **b** In this row, we show the resulting submarine fan morphology by doubling the height of the current (h) and reducing the concentration (**c**) by half to keep equivalent dimensionless value for $Fr_D$ **c** In this row, we increase the mean velocity (U) of the current at the inlet by proportionally increasing the height of the current (h) and concentration of suspended sediments (**c**). We show that while the precise morphology of the simulated fans changes as a result of the variation in the dimensional scales, the overall shape remains similar and shows strong dependence to unique dimensionless values of $Fr_D$ and $p$ of turbidity current at the inlet. All models have identical grid dimensions [35 × 30 km] and color bars denote thickness [m].

assessment and estimation of organic carbon fluxes and reservoirs in the deep ocean.

## Discussion

Defining conditions that lead to the diversity of submarine fan morphologies found on the ocean floor opens the potential for inversion of fan form for paleo-environmental reconstructions and prediction of their future role in the global carbon cycle. We link the complexity of submarine fan form to critical dimensionless parameters that are proxies for the gradients of continental margins and the particle sizes in transport. However, we note these *dimensionless* variables are constructed with important *dimensional* parameters (e.g., current thickness, mean velocity, and diameter of particles in transport). Further, we note that many large fan systems, including the Zaire[4,45], Indus[54], and Bengal fans[13], are found on low gradient margins and house fine-grained deposits. In contrast, many steep margin and sandy systems are small in scale[2,53]. This dimensional correlation between fan size, margin gradient and preserved deposit texture might point to important contributions from the dimensional parameters housed in the $Fr_D$ and $p$. To test the importance of scale on our results, namely the dimensionless regime, we ran sensitivity tests in which a suite of fans were generated (Fig. 8a–c). These runs had

different combinations of input flow thickness (Fig. 8b), velocity (Fig. 8c), and particle sizes, but shared equivalent $Fr_D$ and $p$. The results of these tests support our hypothesis that submarine fan complexity is set by the $Fr_D$ and $p$ of the turbidity currents that formed them. While gradient and particle sizes delivered to a fan are correlated to a fan's size, the complexity of its shape is set by flow and sediment-transport fields. Specifically, $Fr_D$ influences the morphology of a submarine fan by setting thresholds in erosion and deposition in combination with a choice of $p$. A component of the correlation between fan size, gradient, and particle sizes can also be explained from our dimensionless parameters. For example, a flow that is characterized by low $Fr_D$ and $p$ generates a fan surface that is large in scale due to increased advection lengths of the median grain sizes in suspension, compared to a fan that is formed by flows that have a higher $Fr_D$ and $p$. We show this scale difference in the simulated fans of our regime diagram (Fig. 2) where subcritical fans ($Fr_D < 1$) reach up to 40 kms of downstream length as compared to some supercritical fans ($Fr_D > 1$) that are <10 km in downstream length.

Turbidity currents are intermittent phenomena[16,17], with current heads that are often characterized by higher shear stresses than the body of the flows. We explore the sensitivity of submarine fan morphology to intermittent flows (see methods and Fig. 9a, b). We find that

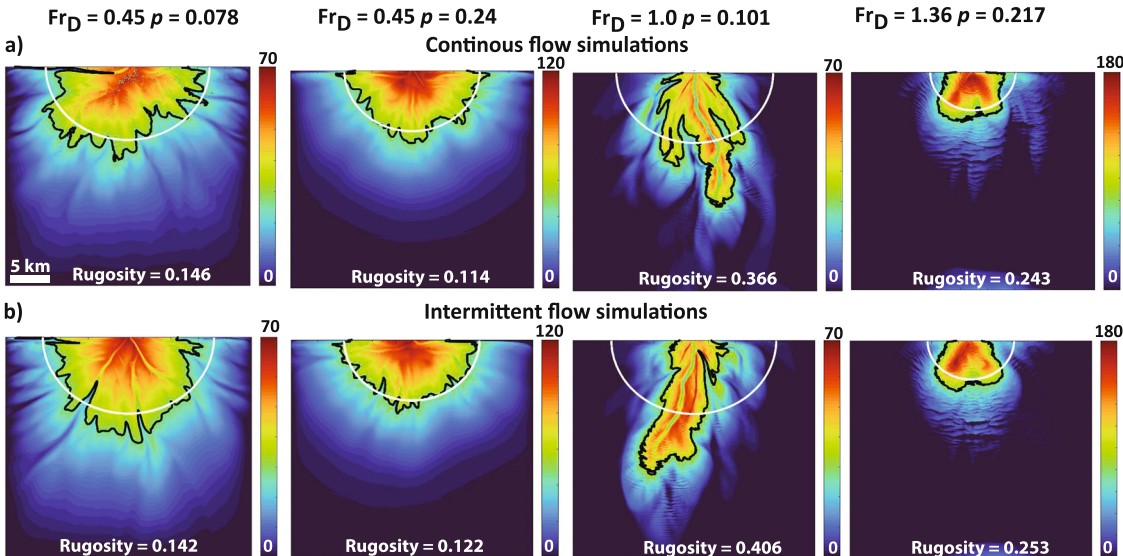

**Fig. 9 | Sensitivity of submarine shape to intermittent turbidity current flow.** **a** Simulated submarine fan morphologies from base case continuous flow conditions as a function of unique combinations of $Fr_D$ and $p$. **b** This row shows simulated submarine fans with intermittent flow conditions. The white semicircle defines a distance from the inlet upstream of which half of the sediments in the model domain are stored. The black contour is computed for a mean elevation along the semicircle. Color bars denote thickness [m]. Note that the relative to the base case simulation the absolute difference in measured rugosity ranges from 2% to at most 10%. Simulation domain is 30 × 30 km.

flow intermittency does not significantly influence fan rugosity, with values changing due to flow intermittency by 2–10%. However, we note that intermittent flows built relatively deeper channels. Intermittent flows introduce relatively more frequent current heads that possess relatively higher shear velocities, which help maintain sediment in suspension. This phenomenon suppresses sedimentation rates within channels and could potentially impact the frequency or magnitude of avulsions on submarine fans and the resulting stratigraphic architecture. We show that flow intermittency appears to be a second order control on fan shape, when compared to the influence of $Fr_D$ and $p$ (Fig. 9a, b & Supplementary Movies 4–6).

Our last sensitivity analysis centers on the role of cohesive sediment on fan shape. This is motivated by studies of other depositional forms, for example deltas, where sediment cohesion has been linked to the rugosity of shorelines[6]. We investigate this by varying the input mud content in the flow and examine how the percent of the total sediment concentration comprised of clay particles influences the absolute magnitude of the resulting morphology metrics (Fig. 10). We find that while changing mud content over a range of 5–60% of the input flux did change the specific placement of channels and lobes, it only produced modest changes in fan rugosity (at most 17% from our base case models). As such, a third axis defined by mud content could be generated for the regime diagram implemented here to account for erosion and deposition dynamics imparted by the cohesive fraction of the sediment concentration. However, this third axis is likely less significant for fan rugosity than the axes defined by $Fr_D$ and $p$.

For turbidity currents to propagate any significant distance, they must generate enough turbulence to enable suspension of sediments[15,55]. We minimize deposition of sediment at the inlet-fan transition by enforcing a constraint on the largest sediment size ($D_{99}$) at the inlet, such that $w_{s,D99} <= u_*$. This constraint leads to emerging trends in sorting, which becomes a parameter dependent on choices of $Fr_D$ and $p$ at the inlet (Supplementary Fig. 1). We find that sorting influences the fraction of a fan surface covered by active channels and the style and rate of channel mobility. Submarine fans constructed by low $Fr_D$ and $p$, coupled with poor sediment sorting, show a higher degree of channelization and development of distributary networks. The development of these complex networks can be linked to the wide

range of settling velocities for the particles in transport, which leads to spatially variable deposition and erosion rates. This generates multiple nucleation sites for channel initiation and the formation of mouth bars that drive channel bifurcation (Fig. 3c). We note that both degree of channelization and the distributary nature of the emergent channels decreases when $Fr_D$ and $p$ increase, as flows become well sorted. In these fans, the incoming flows rapidly become purely depositional, suppressing the growth of complex channel patterns. We note that at extremely high $Fr_D$ values channelization is lost as most sediment bypasses the domain, hindering levee development that confines flows (Fig. 2). Future investigations could explore different criteria to define input grain size distributions, for example, keeping sorting constant across a spectrum of $Fr_D$ and $p$, which may possibly quantify further the influence of sorting on emergent patterns of channel networks on the fan surface.

While the Knapp-Bagnold criteria aids in reducing deposition downstream of the inlet channel, we note that some models (Low $Fr_D$ and high $p$) produce thick deposits near the end of the inlet channel. This results in a steepening of the fan relative to the initial domain slope that was defined through a choice of $Fr_D$ (Fig. 6a). Much of this steepening is associated with deposition that constructs topography necessary to confine flows early in the simulations. However, even after construction of confining topography, models had steeper slopes than defined with the $Fr_D$ closure, which is independent of $p$. This highlights the need for development of relationships to predict bypass slopes that incorporate $Fr_D$, $p$, and sorting within turbidity currents and which could refine our regime diagram. While deposit slope does evolve over the course of the simulations, we note that $Fr_D$ and $p$ do not spatially evolve significantly within the entrance channel (Supplementary Figs. 2–6). This lends support to our argument that $Fr_D$ and $p$ conditions in canyons or relatively immobile channels control the morphology of submarine fans and the properties of the flows in their emergent channels.

Deepwater submarine fans are the ultimate sink in routing systems and their complex morphologies have been described through a hierarchy of landforms[56], which can be linked to a hierarchy of features preserved in their strata[57,58]. Further, recent work suggests that differences in migration rates of features across hierarchical levels

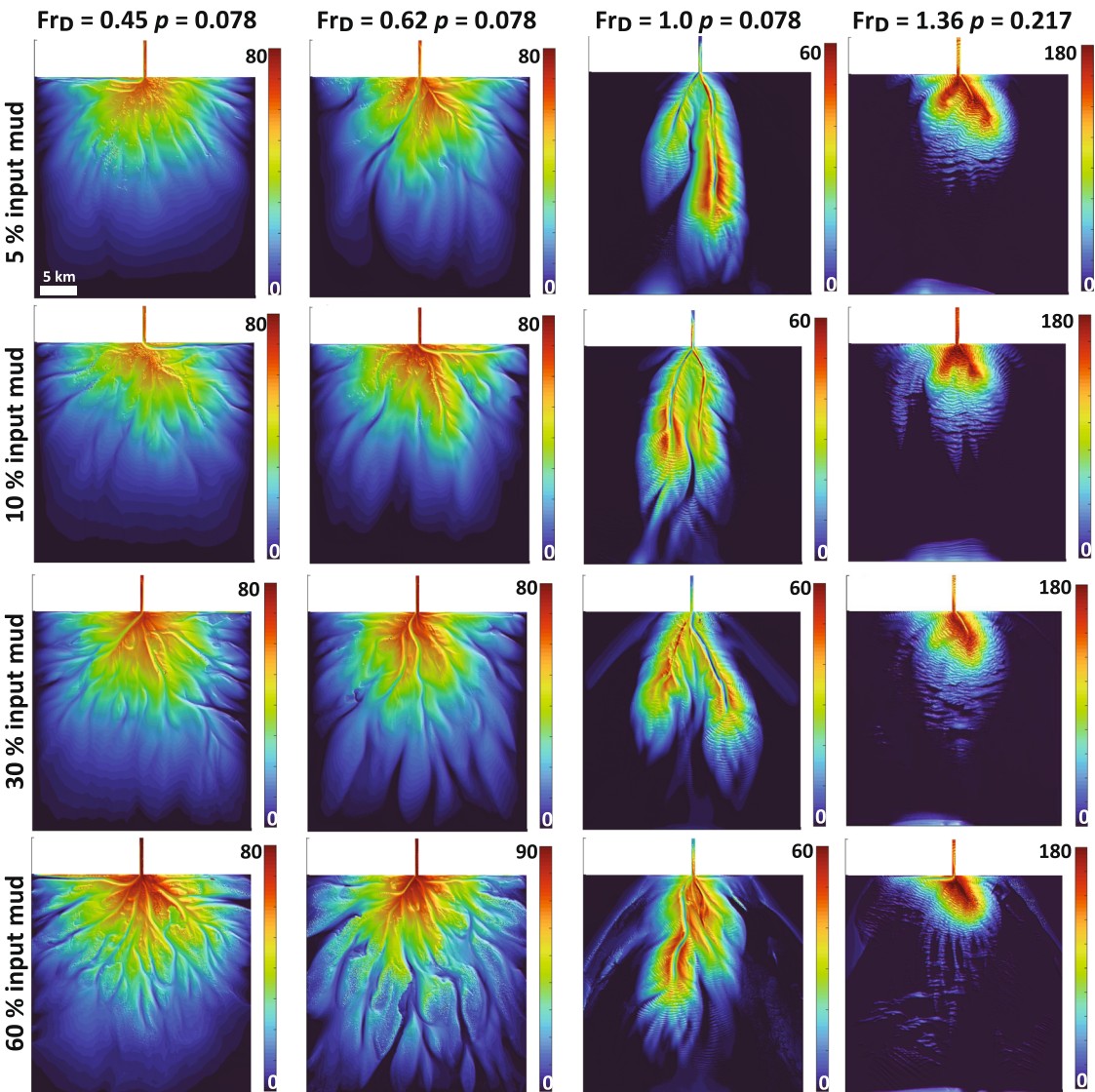

**Fig. 10 | Sensitivity of submarine shape to ranges in mud content.** Matrix shows the influence of cohesion on the resulting fan morphology by increasing input mud content from 5 to 60% at the inlet for a subset fans from the regime diagram. Note the final morphology in these models might be different i.e., placement of channels and lobes on the fan surface, however the overall shape remains similar. Color bar denotes the thickness of the fan at the end of the simulation.

controls the degree to which scales of stratigraphic features (i.e., preserved channel body thickness) resemble common scales of surface features (i.e., channel depth)[59]. This complex stratigraphic hierarchy can develop even under constant environmental forcing, as we show with our model, due to intricate feedback mechanisms between flow and antecedent bed surfaces (autogenics). These autogenic processes and their stratigraphic products makes restoration of signals of paleo-environmental significance challenging[60,61]. The model leveraged in this study produces intricate autogenic behavior and a hierarchy of landforms on the fan surface, all with different migration rates, which vary as a function of $Fr_D$ and $p$ of turbidity currents (Supplementary Movies 1–3). For instance, we show that low gradient fans with relatively low $p$ promotes a distributive network of channels that delivers sediments to many locations on the surface of a fan. The complexity of these networks is not constant over a model run and varies as channels avulse and new channels evolve with different initial conditions. As such, networks can vary in number of active distributaries anywhere between one to many. This intricate and rather dynamic style of autogenic behavior challenges stratigraphers' ability to decipher signals of environmental significance from preserved fan strata, such as from outcrops and seismic volumes. We also show that

high gradient fans crafted by high $Fr_D$ tend to localize deposition in and at the terminus of a single channel network. This style of deposition at the channel-to-lobe transition facilitates choked flow conditions and triggers an avulsion, which results in a shift of locus of deposition on the fan surface. This autogenic configuration reduces the area on the fan surface that is active at any one time. The lack of compensational stacking over these meso-timescales likely imparts significant hiatuses in the fan strata, reducing the temporal completeness of preserved records. The flows that craft these single channel networks also efficiently bypass clay particles that carry sedimentary signals[7]. Combined, periods of depositional stasis on the fan surface and bypass of fines likely limits our ability to leverage high gradient strata for paleo-climatic inversions. The observed complexity in the migration dynamics of channels that emerges as a function of $Fr_D$ and $p$ also likely limits our ability to leverage some reduced complexity models for deciphering signals of climatic and tectonic significance from ordered submarine fan strata. This is due to the limited range of autogenic processes currently produced by some models, which often do not contain rules to describe threshold changes in autogenic processes as a function of flow and sediment-transport conditions[24], as we highlighted in this study.

Results from our dimensionless regime diagram have implications beyond predicting the morphology of submarine fans and the inversion of fan form for paleo-environmental conditions. We show that submarine fans crafted by turbidity currents that are traversing relatively low gradients and transporting fine particles in suspension can efficiently trap more mud compared to fans crafted by turbidity currents traversing higher gradients with relatively coarse particles in suspension (Supplementary Fig. 7). We highlight that flows with subcritical $Fr_D$ and low $p$ favors higher settling rates for sediment that are <62.5 μm in diameter, due to the relatively low shear velocities that characterizes these systems. A high fraction of sediment in the mud size range (<62.5 μm) consists of clays whose mineral surface properties[62] favors adsorption of particulate organic carbon, which can be preserved if rapidly buried to only several cms of depth. This suggest that subcritical fans on low slopes in deepwater settings are favorable spots for sequestering large volumes of organic carbon via increased burial rates of mud supplied by turbidity currents. The higher trapping potential and preservation rates of organic carbon in these fans[13] likely buffers long term atmospheric $CO_2$ levels and can significantly impact global climate change on geological timescales.

We conclude that the remarkable diversity of submarine fan morphology on the modern ocean floor can be generated solely by changing critical dimensionless variables that describe the flow and sediment-transport fields of turbidity currents. This unifying framework ties transport processes to fan form and is defined by the $Fr_D$ and $p$ of turbidity currents. Our results demonstrate the utility of leveraging a process-based forward model, which accounts for all important turbidity current forcing mechanisms, for simulation of complex and geologically realistic submarine fan morphology. Finally, the controls on fan complexity, highlighted in our regime space, will aid inversion of fan form for paleo-environmental conditions and help identify those fans that act as critical reservoirs in Earth's carbon cycle.

## Methods

### Flow-forcing and sediment-transport parameters
In the interest of capturing a range of morphodynamic behaviors within submarine fans that are defined by different slopes and mean grain sizes, we propose a dimensionless regime space. This is achieved by defining the horizontal axis of the regime space as $Fr_D$, which quantifies the ratio of inertial to gravitational forces within a turbidity current. It is calculated as $Fr_D = u/\sqrt{Rgch}$, using depth-average values for important forcing variables within the flow, such as velocity ($u$), concentration ($c$), and height of the current ($h$). We set $c = 5\%$, $h = 5$ m, $R = 1.65$, and $g = 9.8$ m/s$^2$ at the inlet, and therefore, by varying $Fr_D$ we only vary the velocity magnitude at the inlet. We compute for an antecedent slope $S = ((e_w + cd) * Fr_D^2 + 0.5 * e_w)$, as a function of friction at the bed and clear water entrainment from a choice of $Fr_D$, where $e_w$ is the clear water entrainment coefficient, calculated from the $Fr_D$[46], and $cd$ is the coefficient of friction with a fixed value of 0.004. This initial slope is non-erodible. On the vertical axis, we define the $Ro = w_s/(k * u_*)$, which quantifies the ratio of settling velocity of sediments ($w_s$) to shear velocity ($u_*$) in the flow, where $k$ is von Kármán's constant.

### Input grain size distribution and criterion for suspension
We compute the mean settling velocity of sediments $<w_s>$ from a choice of $p$ using the Dietrich[63] method, which then allows us to indirectly compute for the grain sizes in suspension (Supplementary Fig. 1a). We compute a lognormal distribution sampled by 10 quartz sediment sizes with 5 μm clay particles (10% of the total sediment concertation) added to the final sediment mixture. We compute sorting (Supplementary Fig. 1b) for this log normal distribution by leveraging the Shields curve of suspension, such that $D_{99}$ or finer sediment sizes will honor the Bagnold Criterion[15].

### Scaling of the simulation domain
We scale the simulation domain by leveraging a modified version of the advection length, $la = <u> * h_c/w_s$, which describes the minimum horizontal distance over which the mean grain size within a flow can be transported before making contact with the bed[64]. In this formulation, $<u>$ is the depth-average velocity of the flow at the inlet, $h_c$ describes the height of the center of mass within a turbidity current. This height is calculated as, $h_c = h * (0.5 * zc + 0.25(1 - zc))$, where $h$ is the height of the current at the inlet, and $zc$ is the normalized distanced from the bed to the maximum volumetric concentration in the flow, and is set to 0.9 when $Fr_D < 0.5$ and $zc = 0.09 * Fr_D^{-2.8}$ when $Fr_D > 0.5$[65].

### Boundary conditions
The evolution of topography is solved on a flat non-erodible surface with a triangular irregular network (TIN) grid of 50 m × 50 m and 1 m in vertical. For post-processing we interpolate the data onto a Cartesian grid. The entrance channel and upstream boundary walls are closed (and reflect flows), whereas the side and downstream boundaries are open and allow flows to exit the domain. We implement a model stopping condition by fixing the total volume of sediment released into the domain at $1.25 \times 10^{11}$ cubic meters. As a result, subcritical cases require longer computation run-time due to relatively low velocity magnitudes, whereas supercritical simulations take relatively shorter times, given the rapid delivery of the set amount of total sediment volume.

### Simulator
EMstrata uses a four equation model[27] that solves for bedload fluxes[66], erosion rates[67], deposition rates[68], flocculation of clay particles[32,33], and the influence of cohesion on the underlying deposit[31]. Models are solved using a finite volume method with a dynamic time step that is a function of grid spacing, flow velocity, and wave speed. The finite volume method captures discontinuities in the turbidity current flow such as the head of the turbidity current or a hydraulic jump. For our base case scenario of steady currents, once the current is released into the domain it contains a head that propagates through the model domain and eventually exits the simulation domain. This head develops due to sharp spatial gradients in hydrostatic pressure and shear stresses between cells that have near zero velocity or concentration, and those characterizing the current front. The model then simulates the body of the current until a new head originates from a node where the system undergoes an avulsion. The amount of morphodynamic work by the head and the body of the turbidity current is controlled by empirical closures[66–68] that are function of shear stresses within the flow. A more detailed description of the depth-averaged theory, governing equations and analyses of their approximate analytical solutions, and a choice of the numerical solver is published in prior studies[25,26].

### Sensitivity of submarine fan shape to intermittent turbidity current flow
We explore the sensitivity of submarine fan morphology to intermittent flows. We choose four model runs that represented end-member behavior in the regime space (Fig. 9 & Supplementary Movies 4–6). We set simulations by allowing 5 days of active flow discharge followed by 1 day of no flow discharge conditions to allow any remnant current or sediment load to bypass the domain before the next flow event. For these intermittent runs we did not vary any flow forcing conditions from our base case continuous flow simulations. The simulations with intermittent flows also had the same stopping conditions as the base case model runs, where a decided amount of sediment volume was released into the model domain.

## Estimation of Froude and Rouse numbers for modern submarine fan examples

We estimated the $Fr_D$ and $p$ for flows delivered to the Zaire fan using published data of currents in the upstream canyon[52]. We approximate a $Fr_D = 0.13$–$0.21$ based on the measured average current velocities ($U = 0.6$–$1.0$ m/s), the average sediment concentration ($c = 0.020$[16]), and the average height of the current ($h = 64.8$ m). We also estimated the $p = 0.0035$–$0.0057$ for the Zaire fan based on settling velocities ($w_s = 8.62E{-}05$ m/s) of median grain sizes ($d50 = 12$ um) coupled with estimates of shear velocity ($u_* = 0.0379$–$0.0632$ m/s) by assuming a uniform drag co-efficient ($cd = 0.004$) at the bed. We compare this field example to a modeled submarine fan that has a $Fr_D = 0.45$ at the inlet, which is slightly higher than what we estimated for the Zaire system, as this is the smallest $Fr_D$ value modeled in our regime. We do acknowledge that our approximations of $Fr_D$ and $p$ for the Zaire system is an estimate base on limited active turbidity current data. For the intermediate gradient model-field comparison, we use the EastBreaks fan from basin IV of the Brazo-Trinity system offshore Texas. No direct measurements of flows exist for this locality. As such, we estimate $Fr_D$ using Latent-Hyper cube sample of velocity, concentrations, and height of the current. We choose flow variables from this sample space that closely matched the seismic attributes of the EastBreaks fan. The estimates for height = 9 m, velocity = 2.8 m/s, and a concentration of 0.05 yield $Fr_D = 1.05$. We estimate a $p = 0.24$ based on median grain sizes of 200 um from available sediment core data. For the high gradient model-field comparison, we use the Squamish fan system in British Columbia. We estimate the $Fr_D$ ($1.05$–$2.36$) for this system based on active measurements of three flow events with sustain mean body speeds of 3.0 m/s[69] (personal communication with Juan J. Fedele), current height of 10 m (approximated from channel-cross profiles[53], and reasonable choices for concentration of sediments (1–5%). We estimate $p = 0.267$–$0.336$ based on shear velocity ($u_* = 0.150$–$0.1897$ m/s) and settling velocity ($w_s = 0.202$ m/s) of $D_{50} = 225$ um from samples taken in the turbidity current during active flow[70].

## Inversion to median grain sizes in transport

Our method to invert for fan forming flow $Fr_D$ from $p$ and fan rugosity takes advantage of deposit grain size statistics. We highlight a gradient in our regime diagram of a ratio between median particle sizes in transport to the median particle sizes preserved in the fan, $GS_{Ratio}$. This gradient is in the same direction as the observed gradient in fan rugosity as a function of $Fr_D$ and $p$. An empirical relationship was then fit to allow for estimation of $GS_{Ratio}$ from fan rugosity (Fig. 7).

## Data availability

The entire raw dataset (100's of model runs) utilized in this campaign exceeds 100 terabytes. We therefore provide subsampled and processed thickness maps of four model runs that are used to compute for rugosity of simulated submarine fans are available on Figshare at (https://doi.org/10.6084/m9.figshare.21126169). The measured rugosity on this subsampled data may vary slightly from the values reported for the same models in the manuscript due to calculations performed on relatively fewer thickness maps (Supplementary Note 1). However, the authors confirmed that resulting gradients in rugosity across the regime diagram do not change. Therefore, this minimum dataset covers the different corners and gradients of the regime diagram and is sufficient to interpret, verify and extend the research in this manuscript. Additional model outputs could be made available upon reasonable request to the authors. The field data used for comparison to endmember model runs are third party datasets and are available upon request and permission from third party owners (see citations in the manuscript for identity of third party owners).

## Code availability

The code utilized to measure rugosity from thickness maps presented in this paper is available on Figshare (https://doi.org/10.6084/m9.figshare.21131260). The software (EMstrata) utilized in this study for simulating submarine fans is not available for public for propriety reasons. However, the software leverages an established numerical model that has been previously published on (see paper for specific citations). Important forcing parameters explored in this study are detailed in Supplementary Table 1.

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

## Acknowledgements

This study is funded by ExxonMobil Upstream Research Company. We greatly acknowledge members of Process Stratigraphy at ExxonMobil Upstream Research Company, for providing training and access to high performance computing tools to the lead author for generating the dataset utilized in this campaign.

## Author contributions

The authors contributed to this paper as follows, D.C.H, K.M.S, M.S., and A.W. conceived of this study and developed the theoretical framework; A.W. carried out the numerical work with assistance and guidance from M.S.; A.W. processed and interpreted the data; D.C.H., M.S., and K.M.S. reviewed the interpretations; A.W. and K.M.S. wrote the paper with feedback from D.C.H. and M.S.

## Competing interests

The authors declare no competing interests.
