## [Peer Review File · Nature Communications]

A dimensionless framework for predicting submarine fan morphologyReviewer #1 (Remarks to the Author):

Dear Authors and Editor,

I have reviewed the manuscript "A dimensionless framework for predicting submarine fan morphology." I found the manuscript interesting, surely an eventual contribution to the international community. Unfortunately, at the moment, the manuscript is very limited on the discussion of the limitations of the approach and it is lacking very important references. My suggestion is of a Rejection, followed by and Invitation to resubmit after a robust rewrite and refocusing of the identified major issues.

General Comments

I applaud the Authors for proposing a "numerical tank"; this approach is interesting and worth trying but I found the whole introduction and the framing of the project so poorly referenced and substantiated that I could not stop wondering if a better framing was needed. I think the numerical tank is intriguing and helpful to come up with hypotheses to be validated in the field, which is represented by the modern seafloor. The Authors did try to use some modern examples but they used old (almost obsolete) examples. In the last 20 years seafloor mapping has progressed more than ever and speaking about deep-sea fans and showing the Zaire fan with an image from 2002 is not acceptable. Maybe the Authors should look into some recent publication on La Jolla Fan to see the texture of the seafloor they are ignoring. Cherry-picking low-resolution seafloor examples could be fine only if the Authors do a better job at highlighting the limit of their work.

I am in strong agreement when the Authors say: "morphologies influence the fidelity of climate change records housed in their strata"... But, as the Authors say, the strata are where most of those climatic signals are stored not necessarily the fans surfaces (though, I agree they are fundamentally important as a snap shot of what happened recently). Surface morphologies are clearly linked to the strata but the Authors never mentioned how and why. Because I appreciate some of the work presented here, I feel that it would be very important for the Authors to improve the referencing and frame their manuscript better. But let's go with order.

Point to point suggestions:

Line 27: We might "lack a physics-informed theory" (I disagree as I will say in detail later), but a gargantuan volume of data and publications on deep-sea fans could probably better inform the boundary conditions of any "physics-informed" brand new theory. Marine geologists have been working on the deep-sea fans for more than 50 years and we are not that clueless as the Authors seem to think. We should probably start with the seminal work of David Piper and Bill Normark on sandy fans (qualitative, maybe, but excellent!). Even though the manuscript suggested is a 2001 (Piper and Normark, 2001), the contribution is built on more than 30 years of work on deep sea fans, their morphologies and their strata.

Line 31: Citing Heezen et al. is a very nice touch. You are bringing up a classic but there has been a more recent reassessment of this event worth noting (published on Nat Comms so maybe appropriate), Stevenson et al. 2019.

Line 41-45: The Authors (once again) disregard large volume of data and work produced by colleagues and vetted by the community. For instance, Parsons et al. 2002 was followed by a Rowland et al. 2010 with similar setting but very intriguing new insights; hence adding to Parsons 2002. De Leeuw et al should be coupled with another manuscript published few years before (Fildani et al. 2013) because they are incrementally working on something that the Authors describe like unknown. What about the work of Parker's Lab and manuscript such as Cantelli et al. 2011? Line 43 should be completely revised... We do know something about self-forming channels... Please read Cantelli and the many manuscripts published by Parker and his students.

Line 79: A channel is part of a fan. As some of the classic definition from Committee on Fans (ComFan, 1984)... A submarine fan includes channels, lobes, and levees.

Line 124: I believe what the Authors define as cyclical steps have been introduced in the deep-water realm by the Parker's laboratory and called Cyclic Steps -- please call them appropriately. Such Cyclic Steps were never seen in deep-water before 2006 and I invite the Authors to read the manuscript that introduced the concept (Fildani et al. 2006). It could also help to see more recent work dealing with smaller scale cyclic steps finally

imaged with cutting edge technologies on the modern seafloor (Fildani et al 2021). These cyclic steps seem to be everywhere and it is nice to see the Authors obtained them in their runs!

Line 141: I am shocked to see the Authors introducing the concept of hydraulic jumps in deep-water with the Hamilton et al paper from 2015. Please read the classics, once again it could help framing the whole problem better. Mutti and Normark 1987 brought up the process (hypothetically), many others followed on experimental setting (Garcia and Parker seminal work) ... we did not learn this in 2015.

I agree that "a depth-average process-based numerical model to simulate an array of submarine fans and measure rugosity as a proxy for their morphological complexity" is appealing and intriguing but the Authors disregards a lot of the work done by many over the last decades... For instance Line 256-259: The work of Dr. Traer's on sensitivity analyses of turbidity currents is completely disregarded (See Traer et al., 2012, 2015, 2018a, 2018b).

I will stop here. I think the manuscript has potential and I would like to see it published eventually but at the moment it is just an interesting numerical package with no solid link to the real world (yes, the seafloor and related subsurface are the real world). I hope the Authors will have the patience to make the needed improvements and changes. I will be happy to re-review the improved version.

References Suggested:

Cantelli, A., Pirmez, C., Johnson, S. and Parker, G. (2011) Morphodynamic and stratigraphic evolution of self-channelized subaqueous fans emplaced by turbidity currents. *Journal of Sedimentary Research*, 81, 233– 247.

<https://doi.org/10.2110/jsr.2011.20>

Fildani et al., 2006, Channel formation by flow stripping: large-scale scour features along the Monterey East Channel and their relation to sediment waves: *Sedimentology*, v. 53, p. 1265–1287.

Fildani et al., 2021, Exploring a new breadth of cyclic steps on distal submarine fans. *Sedimentology*. 68, 4, 378-1399.

Piper, D.J.W. and Normark, W.R. (2001) Sandy fans - from Amazon to Hueneme and beyond. *AAPG Bulletin*, 85 (8). pp. 1407-1438.

Rowland, J., et al., 2010, A test of submarine leveed channel initiation by deposition alone. *Journal of Sedimentary Research*, 80, 710-727.

Stevenson, C.J., Feldens, P., Georgiopoulou, A. et al. Reconstructing the sediment concentration of a giant submarine gravity flow. *Nat Commun* 9, 2616 (2018).

<https://doi.org/10.1038/s41467-018-05042-6>

Traer, M.M., et al., 2018a, Turbidity Current Dynamics: 1. Model Formulation and Identification of Flow Equilibrium Conditions Resulting from Flow Stripping and Overspill. *J. Geophys. Res. Earth. Surf.*, 123(3), 501-519.

Traer, M.M., et al. 2018b, Turbidity Current Dynamics: 2. Simulating Flow Evolution Toward Equilibrium in Idealized Channels. *J. Geophys. Res. Earth. Surf.*, 123 (3), 520-534.

Traer, M.M., et al., 2015, Simulating depth-averaged, one- dimensional turbidity current dynamics using natural topographies. *J. Geophys. Res. Earth. Surf.*, 120, 1485-1500.

Traer, M.M., et al., 2012, The Sensitivity of Turbidity Currents to Mass and Momentum Exchanges Between These Underflows and Their Surroundings. *J. Geophys. Res. Earth. Surf.*, 117, F01009, 16. doi:10.1029/2011JF001990

Reviewer #2 (Remarks to the Author):

This is a very interesting paper describing how an established numerical model can be used to investigate key dimensionless flow parameter controls on evolution of sedimentary seafloor landscapes, at medium spatial and temporal scales. The manuscript highlights how different seafloor landscapes are the results of key dimensionless parameterisations. Linking these results back to the real world is the key

insight and contribution from this work – I strongly support its publication.

Whilst the paper presents the model used, there is some work needed to highlight its limitations and how these may be reflected in the ultimate results. Further, there remains a key question of the difference between continuous and discrete events which needs to be explored to enable this question to be answered. Finally, the study concludes with an extension to cohesive sediment. I believe that the manuscript would be best served by removing this additional focus and concentrating on the initial hypothesis established.

**Robert Dorrell
University of Hull
13/02/2022**

Comments

L8 I think at this stage with the extensive research in Monterey, Canadian Fjords, Congo and Taiwan over the past 10-20 years that it is unfair to say that there are limited observations. However, I think it is fair to say that the nature of these flows have limited the scope of observations. Please reword accordingly.

L47 As phrased this is misleading “large” numerical simulations, developed over the past 20 years, we developed to provide insight into simplified models of density (and turbidity) current dynamics. Simplified models have extended back over 70 years, see references in Huppert 2006. Could you clarify what is meant here.

L59 After reviewing the methods, and references provided, it is important to note that there are established concerns on the accuracy of simplified depth averaged models in predicting material, momentum and energy fluxes, where vertical structure is not resolved see e.g. Dorrell et al. (2014). (Stratification acts to enhance fluxes, and directly leads to the concept of transcriticality discussed later in the manuscript) Please provide more details, or discuss the relevant limitations, on how the model resolves vertical structure. It should be highlighted that in the manuscript Rouse and Froude number controls are used to define inlet conditions, yet in reality Froude/Rouse number also determine flow structure (Wells & Dorrell, 2021) and thus key fluxes controlling the flow.

L69 Please amend Rouse number defined as Ro to another notation (Ro commonly defines Rossby number).

L77 I assume the advection length is uh/ws ? Please can you state in the manuscript.

L78 It would be appropriate here to introduce the range of background slope of the experimental domain. How does this slope compare to the real-world and where does it lie within the “ignitive” flow regime of Parker et al. (1986). Are flows and morphology produced erosional, analogous to proximal regions of submarine canyons or is the background slope & system representative of canyon-channel systems?

L99 Note on Figure 2 $Fr = 0.148$ rounds up to 0.15 not 0.14.

L112 The images and videos provide qualitative agree with the shape of real-world systems. The models do not however approach the scaling of natural systems. What are the explanations for the limited run-out in the models: current limited understanding (and thus parametrisation of) physical process; timescales of simulation; or other reasons?

L112 Are results presented formed from continuous discharge flows?

This is not the case in the real-world, where multiple discontinuous events occur. There is significant evidence that the heads of these flows play a disproportionate role in material transport (Azpiroz et al. 2017). Assuming continuous discharge flows, it is

unapparent how to fully connect simulations to real-world deposits.

Before phase-space analysis, the manuscript would greatly benefit from addressing some key questions:

> How is the head of the turbidity current resolved during model start up?

> What is the difference between the type (erosional vs aggradational) and amount of morphodynamic work done by the head of the current and the difference between the body and the head?

> A contrast of morphology from a single run of interest (say Fr 1, Rouse number 0.14), which is simulated over 575 days, to results from 115 flows each run over 5 days?

L114 There is theoretical, and field, evidence that (using bulk flow parameters) the sub-super transition occurs below $Fr=1$ (see point above L59, Sumner et al., 2013; and Dorrell et al., 2016 and references therein). Care should be taken when defining supercritical or subcritical flow.

L118 The comparison of low Froude number flows and the Congo canyon is interesting. Does the network of short active channels really match the single long canyon of the modern Zaire. Further, instead of 3a an annotated figure showing sediment deposit thickness (Picot et al., 2016) would be informative to contrast to Figure 3e. Finally, is it possible to make comparison between observations of the Froude number in the Zaire vs model results (Simmons et al., 2021).

L159 Presumably this should be a double integral of deposit thickness over r and $\theta = 0..2\pi$. Alternatively please define B and H as equations.

L168 Is the rugosity index a standard model – if so please provide a reference. Further, could you rephrase the description of this equation for ease of understanding. Where you have n “ x - y locations on the semi circle” but quantify “ x - y locations on the mean thickness contour”.

Would an easier quantification of rugosity be the area enclosed by the mean contour line, within $r(\chi=0.5)$, versus the area enclosed by $r(\chi=0.5)$?

In the methods could you explain what happens as the number of sample points is increased, justifying the number of sample points used?

L171 Suggest use different white symbols, not colours, to enable data to be distinguished in Figure 6.

L172 Why exclude bypass results? These would be interesting, if even to highlight phase change in rugosity.

L214 I do not think it is needed to quote regression to 4-significant figures. There are insufficient data points to justify this.

- Figure 2 – 8 The amount of deposition is high in all models, tens-hundreds of metres in the first few km of simulation. Indeed it is greater than the background slope, which is order few metres over the same distance. Some downdip transects of deposit depth would be very informative. In the Discussion could you please consider to what extent simulations are actually setting their own slope (see e.g. points above on L59 and L78) as a function of inlet conditions, rather than as a function of background slope. What is the final simulated slope versus initial slope? Further, using downdip transects or otherwise, could you comment on the role of deposits in flow blocking in the simulations (Hamilton et al., 2015).

L229 Figure 8 – what are the grid dimensions?

L249 The study of cohesive sediment-laden flows is very interesting. However, I believe it goes beyond the scope of the current paper (analysis on Froude number and Rouse number controls). I would suggest omitting it from the current study, and developing it as a separate contribution.

L258 I don't think the autosuspension criterion $u^* \geq 2U > ghR_{cws}$ is enforced, see point above on Figure 2-8 on excessive deposition near the inlet. Note instead of the U/wsS criteria, this form of the autosuspension criteria (or indeed the TKE produced $>$ buoyancy production + dissipation) may be more applicable for high Fr where entrainment is non-negligible (especially where in such stratified flows dissipation may be assumed large in comparison to Buoyancy production). Further, if the noted form of autosuspension is used, it should at least use the evolved bathymetric not initial background slope. Finally, it is highlighted that the criteria cannot be defined by a strict equality due to the role of dissipation (see also self-accelerating flow condition of Parker et al., 1986).

Indeed the paper would benefit from an analysis of $u^* \geq 2U / ghR_{cws}$ with a figure enabling analysis of model capability to capture autosuspension (noting points above). Where autosuspension is not captured this may explain the limited spatial extent of simulated flows, compared to the real-world. Further, it may be explained by linking back to recent work looking at turbulent mixing in turbidity currents as controls on the autosuspension process (see Wells and Dorrell, 2021 and references therein).

References

- Azpiroz-Zabala, M., Cartigny, M. J., Talling, P. J., Parsons, D. R., Sumner, E. J., Clare, M. A., ... & Pope, E. L. (2017). Newly recognized turbidity current structure can explain prolonged flushing of submarine canyons. *Science advances*, 3(10), e1700200.
- Dorrell, R. M., Darby, S. E., Peakall, J., Sumner, E. J., Parsons, D. R., & Wynn, R. B. (2014). The critical role of stratification in submarine channels: implications for channelization and long runout of flows. *Journal of Geophysical Research: Oceans*, 119(4), 2620-2641.
- Dorrell, R. M., et al. "Flow dynamics and mixing processes in hydraulic jump arrays: Implications for channel-lobe transition zones." *Marine Geology* 381 (2016): 181-193.
- Hamilton, P. B., Strom, K. B., & Hoyal, D. C. (2015). Hydraulic and sediment transport properties of autogenic avulsion cycles on submarine fans with supercritical distributaries. *Journal of Geophysical Research: Earth Surface*, 120(7), 1369-1389.
- Huppert, H. E. (2006). Gravity currents: a personal perspective. *Journal of Fluid Mechanics*, 554, 299-322.
- Picot, M., Droz, L., Marsset, T., Dennielou, B., & Bez, M. (2016). Controls on turbidite sedimentation: Insights from a quantitative approach of submarine channel and lobe architecture (Late Quaternary Congo Fan). *Marine and Petroleum Geology*, 72, 423-446.
- Wells, M. G., & Dorrell, R. M. (2021). Turbulence processes within turbidity currents. *Annual Review of Fluid Mechanics*, 53, 59-83.
- Simmons, S. M., Azpiroz-Zabala, M., Cartigny, M. J. B., Clare, M. A., Cooper, C., Parsons, D. R., ... & Talling, P. J. (2020). Novel acoustic method provides first detailed measurements of sediment concentration structure within submarine turbidity currents. *Journal of Geophysical Research: Oceans*, 125(5), e2019JC015904.
- Sumner, E. J., Peakall, J., Parsons, D. R., Wynn, R. B., Darby, S. E., Dorrell, R. M., ... & White, D. (2013). First direct measurements of hydraulic jumps in an active submarine density current. *Geophysical Research Letters*, 40(22), 5904-5908.

Reviewer #1 (Remarks to the Author):

Dear Authors and Editor,

I have reviewed the manuscript “A dimensionless framework for predicting submarine fan morphology.” I found the manuscript interesting, surely an eventual contribution to the international community. Unfortunately, at the moment, the manuscript is very limited on the discussion of the limitations of the approach and it is lacking very important references. My suggestion is of a Rejection, followed by an Invitation to resubmit after a robust rewrite and refocusing of the identified major issues.

AW: We thank reviewer #1 for reviewing our manuscript. Given comments from both reviewers, we see that we might have left a sense of too much confidence in the model we utilize and much too limited text to acknowledge the limitations of our study. This was not our intention, but we are thankful that it was raised, and we have made a concerted effort to address this in the revised document. Specifically, we have added more details {line 64-76} about the limitations of the depth-average model.

"The depth-averaged approximations used in this model successfully simulate self-formed channels on fan surfaces that are laterally mobile and evolve through time to generate complex submarine fan morphologies. However, the depth averaged approach means the model does not resolve evolution of surface morphologies tied to the vertical flow structure (e.g. ripples, dunes and anti-dunes). Further, this modeling approach utilizes top hat functions for flow concentration and velocity to determine properties of flows that are stripped by partial confinement, which recent work highlights as limitations when modeling the morphodynamics of submarine fans. Specifically, it has been shown that use of self-similar shape functions that describe flow structure, and which vary as a function of the densimetric Froude Number (Fr_D) and Rouse Number (p), can aid estimation of mass and momentum fluxes (Dorrell et al. 2014). These limitations noted, EMstrata captures rich dynamics of turbidity currents and their resulting deposits with reasonable computational costs, allowing investigation of hundreds of models. "

We also expand the scope of the work cited to address some deficiencies noted by reviewer #1 and made changes to introductory text that we outline below.

General Comments

I applaud the Authors for proposing a “numerical tank” ; this approach is interesting and worth trying but I found the whole introduction and the framing of the project so poorly referenced and substantiated that I could not stop wondering if a better framing was needed.

AW: While we like the phrasing of “numerical tank”, which is catchy and a good phrase, we think it’s necessary to highlight that we do not propose a numerical tank in this manuscript. We note and want to emphasize the excellent work to construct EMstrata that has been published on already and which we cite. Rather our aim is to propose and explore a dimensionless approach that helps in simplifying critical forcing parameters for turbidity currents such as the antecedent slope and properties of sediments through inlet flow Froude and Rouse numbers {See lines 10-

14; 113-115}. Next, we do appreciate the review's words about referencing. Our aim was to provide a detailed list of references that support our text, but we acknowledge that this list could be longer and the framing of the intro could do a better job acknowledging the work we are building on. This is where we all lean on reviewers to help us make better documents and this reviewer has certainly helped along this front. Upon reviewer #1's suggestion, we made changes to the text in the introduction part of the manuscript, including buttressing our existing references to better acknowledge the scope of prior work.

I think the numerical tank is intriguing and helpful to come up with hypotheses to be validated in the field, which is represented by the modern seafloor.

AW: Thank you for this comment and acknowledging our numerical approach that quantifies the shapes of different submarine fans represented by the modern seafloor. We agree that a key is to link our hypotheses to field scale observables. We make a start down this path in this manuscript, but certainly there is more work to be done on this front.

The Authors did try to use some modern examples but they used old (almost obsolete) examples. In the last 20 years seafloor mapping has progressed more than ever and speaking about deep-sea fans and showing the Zaire fan with an image from 2002 is not acceptable.

AW: In our manuscript we focus on comparison of end member fan simulations to field examples with comparable gradients to show the efficacy of our modeling approach in generating fan surfaces that display remarkable similarity to their field counterparts {see line 126-170}. For the low gradient (subcritical) submarine fan example we compare our numerical results to the Zaire fan that has forcing similar to one of the model runs {See line 466-477}. While we agree with reviewer #1 about the significant advances in sea floor mapping since the early 2000's, we still lack bathymetric maps in low gradient systems (i.e. Zaire, Congo, Indus, Amazon) of the resolution found on high gradient settings (i.e. La Jolla fan). In addition we would also like to state that given the sheer scale of the Zaire system, and the size of grid cells covering this fan, the bathymetric map is of comparable resolution to some small submarine fan examples like the Squamish or La Jolla.

Maybe the Authors should look into some recent publication on La Jolla Fan to see the texture of the seafloor they are ignoring. Cherry-picking low-resolution seafloor examples could be fine only if the Authors do a better job at highlighting the limit of their work.

AW: We thank the reviewer for their comments. We are aware of the excellent work on the La Jolla fan, specifically the detailed observations of the texture of the modern sea floor. This highlights that we now have several high gradient sites with excellent topographic observations. While we focus our high gradient comparison to a different system, it is not to diminish the work from the La Jolla fan. Here we just want to highlight that in our manuscript we compared end member modeled submarine fans to their field counterparts to highlight the efficacy of our classification scheme in generating seafloor morphology that are represented by the modern seafloor. As stated above, the reference by reviewer #1 to La Jolla fan is an example of a high-

gradient (supercritical) submarine fan. Fildani et al (2021; figure 4) reported a slope of about 0.5° for a longitudinal profile that covers the channel to lobe transition in the La Jolla fan. Based on our approach of computing slope from a choice of Froude number, this slope facilitates supercritical turbidity current flow in the range ($Fr_D = 1.05 - 1.17$). We already compare a high-resolution bathymetric map of a supercritical ($Fr_D = 1.05 - 2.36$) fan example, i.e. the Squamish fan to one of our models with comparable inlet flow forcing conditions. Both La Jolla and the Squamish fan display similar surface morphology, where the fan surface is characterized by a single channel that shows evidence of avulsion and formation of cyclic steps. In addition, the bathymetric surveys of the Squamish fan are of high spatial resolution. As such, our intention was not to cherry pick field examples for comparison to model results but to compare models to similarly forced field examples of submarine fans, with the highest resolution afforded by available datasets. As stated earlier, we still do not have any high-resolution bathymetric maps for low gradient field examples such as the Zaire, Indus, and Amazon submarine fan.

I am in strong agreement when the Authors say: “morphologies influence the fidelity of climate change records housed in their strata” ... But, as the Authors say, the strata are where most of those climatic signals are stored not necessarily the fans surfaces (though, I agree they are fundamentally important as a snap shot of what happened recently). Surface morphologies are clearly linked to the strata but the Authors never mentioned how and why. Because I appreciate some of the work presented here, I feel that it would be very important for the Authors to improve the referencing and frame their manuscript better. But let's go with order.

AW: We appreciate the concerns raised by the reviewer about the importance of the stratigraphic architecture of submarine fans. However, in this manuscript we mainly focus on the surface process and their link to fan shape. While the stratigraphic record is a rich archive of Earth's history, the field of geomorphology has noted for many years that the shape of the Earth's surface not only reflects current system forcings but carries with it information of Earth's history. This holds for both high and low gradient settings. Subsequent work that utilizes the model employed here is exploring the stratigraphic architecture and its connection to observed surface morphologies. However, our focus here is to link forcing conditions to surface morphologies of these systems, similar in a manner to how the coastal geomorphology community has linked sediment properties, and wave and tide climates to the surface expression of deltas (Caldwell and Edmonds, 2014; Nienhuis et al. 2020). We feel that inclusion of work on the strata is beyond the scope of our papers stated goals.

Point to point suggestions:

Line 27: We might “lack a physics-informed theory” (I disagree as I will say in detail later), but a gargantuan volume of data and publications on deep-sea fans could probably better inform the boundary conditions of any “physics-informed” brand new theory. Marine geologists have been working on the deep-sea fans for more than 50 years and we are not that clueless as the Authors seem to think. We should probably start with the seminal work of David Piper and Bill

Normark on sandy fans (qualitative, maybe, but excellent!). Even though the manuscript suggested is a 2001 (Piper and Normark, 2001), the contribution is built on more than 30 years of work on deep sea fans, their morphologies and their strata.

AW: We appreciate the point made by the reviewer. Our intention is not to diminish prior qualitative and semi-quantitative work on submarine fans as qualitative observations of fan shape and strata have inspired models of turbidity currents and quantitative theory for their resulting stratigraphic architecture. Acknowledging this, as we now attempt to with greater referencing, we want to highlight the community's lack of a quantitative and physics informed theory/classification that can simplify the sediment-transport and flow dynamics of turbidity currents, which are responsible for generating submarine fans. For example, early work on fan architecture, including the seminal work of Piper and Normark (2001), generally carries tremendous scatter. The work of Piper and Normark (2001) show a qualitative classification scheme for submarine fans (figure 15), where submarine fans are classified as a function of sediment texture (could roughly be considered as proxy for rouse number in turbidity currents) and initiation mechanisms for turbidity currents (not a proxy for quantifying the flow-forcing conditions within turbidity currents). However, we highlight that both Rouse and Froude state are key in determining fan shape in a quantitative framework. Our novel classification schemes highlight differences in the morphodynamics of turbidity currents. For example, we show that low gradient (subcritical fan $Fr < 1$) have generally low rugosity, preserve higher mud content, are characterized by multiple sinuous distributive channels that fluctuates between one-to-many, preserves a hierarchy of bifurcations, and a relatively complex topographic evolution. On the other hand, submarine fans that are formed on high gradients (supercritical $Fr > 1$) are characterized by high rugosity values, relatively fewer avulsing channels, sandy fabric, and cyclic steps. This adds a predictive ability that did not exist before this work and we are able show the validity of our classification approach by comparing observation from modeled fans to field example that share remarkable similarity (Figure 3-6).

Line 31: Citing Heezen et al. is a very nice touch. You are bringing up a classic but there has been a more recent reassessment of this event worth noting (published on Nat Comms so maybe appropriate), Stevenson et al. 2019.

AW: We thank the reviewer for suggesting this reference. We now cite Stevenson et al (2019) in our manuscript.

Line 41-45: The Authors (once again) disregard large volume of data and work produced by colleagues and vetted by the community. For instance, Parsons et al. 2002 was follow by a Rowland et al. 2010 with similar setting but very intriguing new insights; hence adding to Parsons 2002. De Leeuw et al should be coupled with another manuscript published few years before (Fildani et al. 2013) because they are incrementally working on something that the Authors describe like unknown. What about the work of Parker's Lab and manuscript such as Cantelli et al. 2011? Line 43 should be completely revise... We do know something about self-

forming channels... Please read Cantelli and the many manuscripts published by Parker and his students.

AW: We appreciate the comments raised by reviewer #1, and we are aware of the experimental work referred to. In fact, some of the authors on this paper are physical experimentalist and would love to be able to tout the ability of physical experiments to explore these problems. However, we find that work mentioned by reviewer #1 never produced self-formed channels via flows that were undeniably turbidity currents and that display sufficient mobility (continuous lateral migration and avulsions) to form a fully dynamic fan surface. Experimental work by Parsons et al. (2002) and Rowland et al. (2010) failed to achieve morphodynamic conditions necessary for formation of self-channelized submarine fans (acknowledging the assertion of Rowland et al. 2010 that erosion must be necessary for channel initiation). While we also applaud the work of De Leeuw et al. (2016), specifically how channel initiation is quantitatively linked to fan roughness scales and suspension criteria, the fact remains that this study achieved a single self-formed channel with a length that was less than 10 times its width and that never avulsed. As such even this seminal work is far from generating a fully channelized submarine fan. Cantelli et al.'s (2011) work generated self-formed channels from flows that approached 10% sediment concentration, which is much greater than concentrations reported for natural turbidity currents (1-5%). Further, given the scale of the flows in these experiments, low flow velocities, and high concentrations, it is debatable whether the channel forming flows can be termed as fully turbulent turbidity currents. This is even acknowledged by Cantelli and his co-authors in their manuscripts (they note that their flows might best be described as dilute mudflows, where mudflows have "discernable yield strength"). Therefore, it is still an existing challenge for experimentalist to generate self-channelized submarine fans constructed by fully turbulent flows that transport most of their sediments in suspension in a laboratory setting. As such, we feel that for our study, we place our work in appropriate context.

Line 79: A channel is part of a fan. As some of the classic definition from Committee on Fans (ComFan, 1984)... A submarine fan includes channels, lobes, and levees.

AW: Thank you for your comment. We now state the following {lines 93-95,}

"Further, the setup of our numerical experiments is designed to explore net depositional submarine fans downstream of canyons or deposits that develop downslope of relatively immobile feeder channels."

Line 124: I believe what the Authors define as cyclical steps have been introduced in the deep-water realm by the Parker' s laboratory and called Cyclic Steps -- please call them appropriately. Such Cyclic Steps were never seen in deep-water before 2006 and I invite the Authors to read the manuscript that introduced the concept (Fildani et al. 2006). It could also help to see more recent work dealing with smaller scale cyclic steps finally imaged with cutting edge technologies on the modern seafloor (Fildani et al 2021). These cyclic steps seem to be everywhere and it is nice to see the Authors obtained them in their runs!

AW: Thank you for your comment. This change has been made from cyclical steps to cyclic steps in the manuscript. We also cite Fildani et al (2006).

Line 141: I am shocked to see the Authors introducing the concept of hydraulic jumps in deep-water with the Hamilton et al paper from 2015. Please read the classics, once again it could help framing the whole problem better. Mutti and Normark 1987 brought up the process (hypothetically), many others followed on experimental setting (Garcia and Parker seminal work) ... we did not learn this in 2015.

AW: We thank the reviewer for this suggestion. We have added Mutti and Normark (1987) and Garcia and Parker (1989).

I agree that “a depth-average process-based numerical model to simulate an array of submarine fans and measure rugosity as a proxy for their morphological complexity” is appealing and intriguing but the Authors disregards a lot of the work done by many over the last decades... For instance Line 256-259: The work of Dr. Traer’ s on sensitivity analyses of turbidity currents is completely disregarded (See Traer et al., 2012, 2015, 2018a, 2018b).

AW: We thank the reviewer for this comment. However, we are confused about the placement of the comment. Lines 293-295 reference text where we detail a suspension criterion that we enforce for our input sediment. This criteria dates from papers in the 1930’s and 1960’s. These are not modeling papers, but solely detail conditions for sediment suspension. We are simply stating that sediment in a flow must be able to be suspended by the flow (turbidity currents need to have sediment in suspension to drive the flow). This condition does not deal with measuring fan rugosity or morphology. While the work of Dr. Traer and others have advanced our understanding of turbidity currents and how to model them, this work does not have any bearing on the condition we discuss in these lines. However, we do add text earlier in the manuscript to acknowledge the vast literature that has explored modeling of turbidity currents in the past (with an array of model types) { see line 61 }.

I will stop here. I think the manuscript has potential and I would like to see it published eventually but at the moment it is just an interesting numerical package with no solid link to the real world (yes, the seafloor and related subsurface are the real world). I hope the Authors will have the patience to make the needed improvements and changes. I will be happy to re-review the improved version.

AW: We thank the reviewer for this comment and their review. We hope the revisions included in the resubmission, including our work to estimate the Fr_D and p of the field scale examples given (and comparison to similar Fr_D and p models) provides the link to the real world that the reviewer #1 feels was missing in the initial submission.

Reviewer #2 (Remarks to the Author):

This is a very interesting paper describing how an established numerical model can be used to

investigate key dimensionless flow parameter controls on evolution of sedimentary seafloor landscapes, at medium spatial and temporal scales. The manuscript highlights how different seafloor landscapes are the results of key dimensionless parameterizations. Linking these results back to the real world is the key insight and contribution from this work - I strongly support its publication.

AW: We thank the reviewer for this comment and the overarching constructive nature of his review.

Whilst the paper presents the model used, there is some work needed to highlight its limitations and how these may be reflected in the ultimate results. Further, there remains a key question of the difference between continuous and discrete events which needs to be explored to enable this question to be answered. Finally, the study concludes with an extension to cohesive sediment. I believe that the manuscript would be best served by removing this additional focus and concentrating on the initial hypothesis established.

AW: We thank the reviewer for this comment. We now include the following text {line 64-76} in the manuscript that highlights the limitation of depth-average theory for sedimentology and stratigraphy of submarine fans.

“The depth-averaged approximations used in this model successfully simulate self-formed channels on fan surfaces that are laterally mobile and avulse through time to generate complex submarine fan morphologies. However, the depth averaged approach means the model does not resolve evolution of surface morphologies tied to the vertical flow structure (e.g. ripples, dunes and anti-dunes). Further, this modeling approach utilizes top hat functions for flow concentration and velocity to determine properties of flows that are stripped by partial confinement, which recent work highlights as limitations when modeling the morphodynamics of submarine fans. Specifically, it has been shown that use of self-similar shape functions that describe flow structure, and which vary as a function of the densimetric Froude Number (Fr_D) and Rouse Number (p), can aid estimation of mass and momentum fluxes (Dorrell et al. 2014). These limitations noted, EMstrata captures rich dynamics of turbidity currents and their resulting deposits with reasonable computational costs, allowing investigation of hundreds of models.”

Furthermore, upon the reviewer suggestion, we have expanded on our sensitivity analyses to explore the influence of the intermittency of the flow on the resulting submarine fan morphology. We simulated four cases from our regime space that capture the end member behaviors {line 275 – 280, line 457 – 465, Figure 9}. While flow intermittency (and percent of time experiencing a flow head vs. body) exert a second order control on fan shape, we show that the Froude and Rouse number of turbidity currents exert a stronger first order control on submarine fan shape.

Robert Dorrell
University of Hull

13/02/2022

Comments

L8 I think at this stage with the extensive research in Monterey, Canadian Fjords, Congo and Taiwan over the past 10-20 years that it is unfair to say that there are limited observations. However, I think it is fair to say that the nature of these flows have limited the scope of observations. Please reword accordingly.

AW: We thank the reviewer for this comment. We have now changed the language to “Observation of active turbidity currents at field scale offers a limited scope of observation which challenges development of theory that links flow dynamics to the morphology of submarine fans” {lines 8 - 10}.

L47 As phrased this is misleading “large” numerical simulations, developed over the past 20 years, we developed to provide insight into simplified models of density (and turbidity) current dynamics. Simplified models have extended back over 70 years, see references in Huppert 2006. Could you clarify what is meant here.

AW: We thank the reviewer for this comment. To clarify, we state that modeling of turbidity currents has followed several tracks over the last few decades, some of which carry large computational cost and limited temporal scale of exploration. Other rule based models avoid large computational costs by reducing the complexity of the turbidity current models and the resultant morphologies of submarine fans. In our manuscript we follow an intermediate path (a depth-average approach) that carry reasonable computational cost and solves for important flow and sediment-transport parameters of a turbidity current interacting with antecedent surface. To address this comment we have added the following lines {line 46 – 56}:

“These challenges have influenced the development of forward numerical models of turbidity currents and their resulting submarine fans. Some of these models carry large computational costs to solve Navier-Stokes equations for complex and fully turbulent turbidity currents. On the other side of the spectrum are rules based reduced complexity models, some of which have been around for decades but are being refined through field measurements and distillation of observations from the high fidelity models²⁶⁻²⁸. While these reduced complexity models generate realistic gross scale compensational stacking patterns and lobe-scale geometries over an antecedent surface, they did not account for erosion of the underlying deposits. The lack of spatial and temporal variations in deposition and erosion on the fan surface limit their applicability to unchannelized and purely depositional submarine fans.”

L59 After reviewing the methods, and references provided, it is important to note that there are established concerns on the accuracy of simplified depth averaged models in predicting material, momentum and energy fluxes, where vertical structure is not resolved see e.g. Dorrell et al. (2014). (Stratification acts to enhance fluxes, and directly leads to the concept of transcriticality discussed later in the manuscript) Please provide more details, or discuss the relevant limitations,

on how the model resolves vertical structure. It should be highlighted that in the manuscript Rouse and Froude number controls are used to define inlet conditions, yet in reality Froude/Rouse number also determine flow structure (Wells & Dorrell, 2021) and thus key fluxes controlling the flow.

AW: We thank the reviewer for this comment. As noted above, we have added the following text to explain limitations to our approach {lines 64-76}.

“The depth-averaged approximations used in this model successfully simulate self-formed channels on fan surfaces that are laterally mobile and avulse through time to generate complex submarine fan morphologies. However, the depth averaged approach means the model does not resolve evolution of surface morphologies tied to the vertical flow structure (e.g. ripples, dunes and anti-dunes). Further, this modeling approach utilizes top hat functions for flow concentration and velocity to determine properties of flows that are stripped by partial confinement, which recent work highlights as limitations when modeling the morphodynamics of submarine fans. Specifically, Dorrell et al. (2014) show that use of self-similar shape functions that describe flow structure, and which vary as a function of Fr_D and p , can aid estimation of mass and momentum fluxes. These limitations noted, EMstrata captures rich dynamics of turbidity currents and their resulting deposits with reasonable computational costs, allowing investigation of hundreds of models.”

L69 Please amend Rouse number defined as Ro to another notation (Ro commonly defines Rossby number).

AW: We thank the reviewer for this comment. We now change the notation from Ro to p .

L77 I assume the advection length is uh/ws ? Please can you state in the manuscript.

AW: We thank the reviewer for this comment. This parameter is defined in our method section (line 424).

L78 It would be appropriate here to introduce the range of background slope of the experimental domain. How does this slope compare to the real-world and where does it lie within the “ignitive” flow regime of Parker et al. (1986). Are flows and morphology produced erosional, analogous to proximal regions of submarine canyons or is the background slope & system representative of canyon-channel systems?

AW: We thank the reviewer for this comment. The background slope for our simulations range from 0.065° to 3.63° . These slopes are now shown on the upper horizontal axis of figure 2, 6, and table 1. We emphasize here two points: 1) While the range of slopes explored in our regime diagram span those found on continental slopes, we do not pick slopes to model to simulate specific field sites. Rather, the initial domain slope is a dependent variable that is set by a choice of the Fr_D we wish to model, a clear water entrainment coefficient, and a drag coefficient. 2) Flows within EMstrata can ignite and this ignition falls within a similar range of slopes as

defined in Parker et al. 1986, with slight variations due to differences in some closure relations used in EMStrata vs. Parker et al. 1986.

To make clear the range of slopes explored, we have also added the following text {lines 90-93}:

“The initial surface slope is a dependent parameter defined by a choice of Fr_D , clear water entrainment and drag (see methods). The range of Fr_D explored resulted in models with initial slopes between 0.065° to 3.63° , similar to slopes observed on continental margins (Pratson and Haxby 1996).”

Finally, while flows can erode previously deposited sediment, we focus on conditions that range from net depositional to bypass in our analysis. Specifically, our initial condition includes a surface that is non-erosive and as such we cannot and do not explore net erosional conditions. To clarify this, we have added text to this section that states we focus on net depositional settings and within the methods section we now note that the initial slope is non-erodible.

L99 Note on Figure 2 $Fr = 0.148$ rounds up to 0.15 not 0.14.

AW: We thank the reviewer for this comment. This change has been made.

L112 The images and videos provide qualitative agreement with the shape of real-world systems. The models do not however approach the scaling of natural systems. What are the explanations for the limited run-out in the models: current limited understanding (and thus parametrization of) physical process; timescales of simulation; or other reasons?

AW: We thank the reviewer for this comment. First, the input flow heights to the model domain were only 5m (and 10m for some of the sensitivity tests). As flow height increases the length of the system would also increase. Further, we imposed a stopping condition on our runs. This was set to a specific amount of sediment volume discharged to the domain. If we had run our models for a longer period of time the fan size would have grown (but according to our analysis, the rugosity would stay approximately constant during this growth). Prior use of this model by co-authors has been limited to computational domains less than 100 km, only due to computational time limitations. These models, if run long enough, do eventually produce confinement that allows channels to develop over the full 100 km reach. As such, it seems that long runout in these models might be aided by different parameterization of flow stripping processes, but it does not require different assumptions than utilized in EMStrata. Choices on flow height and sediment discharged into the domain in our study were made to allow development of hundreds of complex fans with rugosity values that each reached dynamic steady state, but with small enough currents and low enough total discharges to make computational costs manageable.

In the text we address this with the following statement {lines 134-140}:

“We note that the absolute scales of some of our simulated fans are small relative to field scale systems, particularly the subcritical end members. This is purely a consequence of the input height of our currents and the model stopping condition, which was picked to limit computational costs for the hundreds of runs we perform. Thus, our goal is not to simulate any

particular system, but rather to compare the general morphological complexity of field scale fans with modeled fans that share similar forcings as defined by Fr_D and p .”

L112 Are results presented formed from continuous discharge flows?

This is not the case in the real-world, where multiple discontinuous events occur. There is significant evidence that the heads of these flows play a disproportionate role in material transport (Azpiroz et al. 2017). Assuming continuous discharge flows, it is unapparent how to fully connect simulations to real-world deposits.

Before phase-space analysis, the manuscript would greatly benefit from addressing some key questions:

> How is the head of the turbidity current resolved during model start up?

AW: We thank the reviewer for this comment. We use a finite volume method that captures discontinuities in the flows such as the head of the turbidity current or a hydraulic jump. For our base case scenario of steady currents, once the current is released into the domain it contains a head that propagates through the model domain and eventually exits the simulation domain. However, even in these steady input experiments, multiple turbidity current heads are generated, due to internal dynamics, for example avulsions. When an avulsion event occurs, flow is diverted rapidly into cells that previously were quiescent. As a consequence, a sharp hydrostatic pressure gradient and shear stress gradient is present. This results in more vigorous sediment transport conditions along the front and a thicker current at the front. We have added text to the methods section {lines 441-456}, specifically the subsection titled “Simulator” to clarify this point.

> What is the difference between the type (erosional vs aggradational) and amount of morphodynamic work done by the head of the current and the difference between the body and the head?

AW: We thank the reviewer for this comment. The amount of morphodynamic work by the head and the body of the current is controlled by empirical closures that are function of shear stresses within the flow. To further clarify, the erosion and deposition dynamics are controlled by empirical closures where shear stress is the primary control. As a consequence differences in the imposed shear stress between the head and body controls the amount of morphodynamic work. We have added text to the methods section {lines 441-456}, specifically the subsection titled “Simulator” to clarify this point and the point raised in the prior comment.

> A contrast of morphology from a single run of interest (say $Fr = 1$, Rouse number 0.14), which is simulated over 575 days, to results from 115 flows each run over 5 days?

AW: We thank the reviewer for this comment. It is a most excellent idea! We ran four different models that captures the end member behavior of the regime space with intermittent flows. Active flows were simulated for 5 days following a day of no discharge conditions to allow any remnant current or sediment load to bypass the domain before the next flow event. We want to note that we did not vary any forcing conditions from our base case steady flow simulations except for introducing intermittent flows. The simulations with intermittent flows had similar

stopping conditions as the base case model runs, where a decided amount of sediment volume was released into the model domain. We measured rugosity for these model runs and found differences between 2-10% changes relative to our base cases for measured rugosity values. We emphasize, however, that the bulk trends remain the same for gradients in Froude and Rouse numbers. See added text with more detail about this in the method section titled “Sensitivity of submarine fan shape to intermittent turbidity current flow” {lines 457-465} accompanied with texts in the discussion section {lines 275-280} and a figure in the supplementary section (Figure 9).

L114 There is theoretical, and field, evidence that (using bulk flow parameters) the sub- super transition occurs below $Fr=1$ (see point above L59, Sumner et al., 2013; and Dorrell et al., 2016 and references therein). Care should be taken when defining supercritical or subcritical flow.

AW: We thank the reviewer for this comment. We have loosened our language appropriately and now state that the sub-super transition occurs at $Fr_D \sim 1$. Our goal here is not to explore any specific hard values of Fr_D or p . Rather, the motivation of the entire study is to explore gradients in a phase space.

L118 The comparison of low Froude number flows and the Congo canyon is interesting. Does the network of short active channels really match the single long canyon of the modern Zaire.

AW: We thank the reviewer for this comment. As we emphasize the goal of our studies is not to model any specific system or generate long run out distances (100's of km) that are characteristic of turbidity currents observed on low gradients like the Congo system (Azpiroz et al. 2017). Rather, we explore the influence of Fr_D and p of the turbidity current on submarine fan shape in a dimensionless framework. We acknowledge that the length of self-formed channels in our subcritical fan simulations only extends downstream up to 40-50 km, and that these scales are not comparable to the Congo system which measures to a downstream length of about 800 km. The length of our channels are the result of the choices we made for height of the current (5m at the inlet in our simulations, where as in the Congo canyon an average height of 62 m is reported Azpiroz et al. 2017) and stopping conditions that yielded reasonable computational cost. As flow height increases, the length of the system would also increase as function of advection lengths of grains in transport. To highlight the computational cost for running one subcritical model, we utilize 40 nodes (equivalent to the compute power of 400 laptops with quad-core processors (>3.0 HZ) for 72 hours of run time that yielded 5 terabyte worth of data.

Further, instead of 3a an annotated figure showing sediment deposit thickness (Picot et al., 2016) would be informative to contrast to Figure 3e.

AW: We now compare a deposit thickness map of the active Zaire fan to our modeled submarine fan. The deposit thickness maps for the Zaire are generated using the ZaiAngo surveys acquired in the early 2000a (Babonneau et al. 2010). The seismic coverage and available sediment cores allowed detailed mapping of stratigraphic thickness of the modern part of the Zaire fan system.

Finally, is it possible to make comparison between observations of the Froude number in the Zaire vs model results (Simmons et al., 2021).

AW: We thank the reviewer for this comment. We estimated the Fr_D and p for the input to the Zaire fan using published data from the Zaire Canyon (Azpiroz et al. 2017 and Simmons et al. 2020). We estimate a We approximate a $Fr_D = 0.13 - 0.21$ based on the measured average current velocities ($U = 0.6 - 1.0$ m/s), the average sediment concentration ($c = 0.020$, azpiroz et al. 2017), and the average height of the current thickness ($h = 64.8$ m). We also estimated the $p = 0.0035 - 0.0057$ for the Zaire fan based on settling velocities ($w_s = 8.62E-05$ m/s) of median grain sizes ($d_{50} = 12$ μ m) coupled with estimates of shear velocity ($u_* = 0.0379$ m/s) by assuming a uniform drag co-efficient ($cd = 0.004$) at the bed. We compare this field example to a modeled submarine fan that has a $Fr_D = 0.45$ at the inlet, which is slightly higher than what we estimated for the Zaire system, as this is the smallest Fr_D value modeled in our regime. We do acknowledge that our approximations of Fr_D and p for the Zaire system is an estimate base on limited active turbidity current data. These measurements could get refined in the future upon collection or availability of more data on active flow and grain size measurements. To better connect our end member simulation results to examples from the modern seafloor we now add a section in the methods text titled “Estimation of Froude and Rouse numbers for modern submarine fan examples” {lines 466-491 }

L159 Presumably this should be a double integral of deposit thickness over r and $\theta = 0..Pi$. Alternatively please define B and H as equations.

AW: We thank the reviewer for this comment. This form of the equation was originally published by Paola and Martin (2012) and Strong et al et al. (2005) and we follow this form. It is simply an integral of the mass stored upstream of a distance from the basin inlet, R . Given our initial geometry, flow is able to radially spread, and as such B changes as a function of R (as does deposit thickness at R).

L168 Is the rugosity index a standard model - if so please provide a reference. Further, could you rephrase the description of this equation for ease of understanding. Where you have n “ x - y locations on the semi circle” but quantify “ x - y locations on the mean thickness contour” . Would an easier quantification of rugosity be the area enclosed by the mean contour line, within $r(\chi=0.5)$, versus the area enclosed by $r(\chi=0.5)$? In the methods could you explain what happens as the number of sample points is increased, justifying the number of sample points used?

AW: We thank the reviewer for this comment. We believe that the method proposed by the reviewer is of similar complexity to the method we utilize, as in order to define an area inbound of the $\chi = 0.5$ contour, you would need to define the contour at a large number of spatial locations. Similar implementation was also achieved by Yu et al. (2017) and showed this approach is grid node and density independent.

L171 Suggest use different white symbols, not colors, to enable data to be distinguished in Figure 6.

AW: We thank the reviewer for this comment. This change has been made.

L172 Why exclude bypass results? These would be interesting, if even to highlight phase change in rugosity.

AW: We thank the reviewer for this comment. Models with excessive bypass did not produce any significant deposits so we exclude such runs from the measurement of rugosity metric. Further, it is possible that some of these runs might have eroded into the initial slope if we had allowed erosion into the initial substrate. Given this, and the differences in fan form that might have resulted from incision into the initial state, we think it is best to exclude these from our presentation.

L214 I do not think it is needed to quote regression to 4-significant figures. There are insufficient data points to justify this.

AW: We thank the reviewer for this comment. We now quote regression to 2-significant figures.

- Figure2 - 8 The amount of deposition is high in all models, tens-hundreds of metres in the first few km of simulation. Indeed it is greater than the background slope, which is order few metres over the same distance. Some downdip transects of deposit depth would be very informative. In the Discussion could you please consider to what extent simulations are actually setting their own slope (see e.g. points above on L59 and L78) as a function of inlet conditions, rather than as a function of background slope. What is the final simulated slope versus initial slope? Further, using downdip transects or otherwise, could you comment on the role of deposits in flow blocking in the simulations (Hamilton et al., 2015).

AW: We thank the reviewer for this comment. Here we highlight the following: Our choice of Fr_D informs our choice of slope, however, in our framework slope is independent of p . In essence, this highlights a problem for our community to tackle. At present, we are unaware of any relationship that allows for estimation of an equilibrium slope linked to both Fr_D and p that would result in pure bypass conditions for turbidity currents. Further, the relationship between Fr_D and slope that we utilize only accounts for downstream changes in density flux through clear water entrainment. This generates a problem for currents composed of a distribution of grain sizes, as coarse material might fallout, resulting in downstream gradients in excess density flux. So, at present we lack a relationship between Fr_D and S that holds for a) all Rouse numbers and b) a full distribution of sizes in suspension. The relationship we use is possibly the best available for us at present, but again, more work should be done on this front.

The above being stated, we highlight the following. We measured Fr_D and p at the top of our entrance channel (first cells in the model domain) and at the end of the 5 km entrance channel for four model runs that represents the endmember behavior of the regime space. This represents a propagation distance that is 1000x our initial current height of 5 m. We see minimal changes in

Fr_D and p over this stretch (at most 15% change in Fr_D and 21 % change in p). Further, the emergent channels downstream of the loss of confinement share similar Fr_D and p as our entrance channel, commonly within 10% difference of the inlet channel flow. This supports our assumption that the conditions we are forcing the model with are maintained at the loss of confinement and control the emergent shape of the fans and the flow characteristics within the channels constructing the fans.

However, the reviewer is correct that a general steepening of fans does occur from the beginning to end of simulations. A portion of this steepening can be linked to high proximal deposition early in model runs when topography has yet to develop to confine flows. This early lack of confinement results in flows that rapidly decelerate due to radial spreading. We now indicate the initial and final topographic slopes for the four fans that we run sensitivity analysis on throughout the document in figure 6. In addition, to address this very good question/point, we have added the following text in the revised manuscript {lines 314-327}:

“While the Knapp-Bagnold criteria aids in reducing deposition downstream of the inlet channel, we note that in all models that produce deposits the greatest deposition is found near the end of the inlet channel. This results in a steepening of the fan relative to the initial domain slope that was defined through a choice of Fr_D (Fig. 6). Much of this steepening is associated with deposition that constructs topography necessary to confine flows early in the simulations. However, even after construction of confining topography, models had steeper slopes than defined with the Fr_D closure, which is independent of p . This highlights the need for development of relationships to predict bypass slopes that incorporate Fr_D , p , and sorting within turbidity currents and which could refine our regime diagram. While deposit slope does evolve over the course of the simulations, we note that Fr_D and p do not spatially evolve significantly within the entrance channel and these parameters stay approximately constant in emergent channels. This lends support to our argument that Fr_D and p conditions in canyons or relatively immobile channels control the morphology of submarine fans and the properties of the flows in their emergent channels.”

L229 Figure 8 - what are the grid dimensions?

AW: We thank the reviewer for this comment. We provided a scale bar to highlight domain size but we now specify the model dimensions (35 x 30 km) for more clarity.

L249 The study of cohesive sediment-laden flows is very interesting. However, I believe it goes beyond the scope of the current paper (analysis on Froude number and Rouse number controls). I would suggest omitting it from the current study, and developing it as a separate contribution.

AW: We thank the reviewer for this comment, we do think that cohesive sediment is an obvious parameter to explore in our model, and as such decided to keep the text and figure.

L258 I don't think the autosuspension criterion $u^*^2 U > ghRcws$ is enforced, see point above on Figure 2-8 on excessive deposition near the inlet. Note instead of the U/wsS criteria, this form of the autosuspension criteria (or indeed the TKE produced $>$ buoyancy production + dissipation)

may be more applicable for high Fr where entrainment is non-negligible (especially where in such stratified flows dissipation may be assumed large in comparison to Buoyancy production). Further, if the noted form of autosuspension is used, it should at least use the evolved bathymetric not initial background slope. Finally, it is highlighted that the criteria cannot be defined by a strict equality due to the role of dissipation (see also self-accelerating flow condition of Parker et al., 1986).

Indeed the paper would benefit from an analysis of u^*^2U/ghR_{cws} with a figure enabling analysis of model capability to capture autosuspension (noting points above). Where autosuspension is not captured this may explain the limited spatial extent of simulated flows, compared to the real-world. Further, it may be explained by linking back to recent work looking at turbulent mixing in turbidity currents as controls on the autosuspension process (see Wells and Dorrell, 2021 and references therein).

AW: We thank the reviewer for this comment. We would like to clarify that we do not implement an autosuspension criterion and we acknowledge that we used this term somewhat loosely in the initial draft. Here we highlight that for a given Rouse number decisions have to be made to define a log normal distribution of sediment to supply the current with. To define a log normal distribution of sediments at the inlet, we enforce a constraint, such that an optimal sorting value can be defined given a mean of the distribution. We achieve this by enforcing a constraint on the largest (D_{99}) sediment size at the inlet. This constraint is that $W_{s,D99} \leq u^*$, which enables a selection of sorting based off a defined mean for the log normal distribution of sediments at the inlet. This constraint helps us in fully defining a log normal distribution of sediment for a given p and minimizes proximal deposition in the simulation. Earlier efforts in setting up this regime diagram without this constraint lead to excessive deposition at the inlet-fan transition and did not generate any unique shapes, which now currently emerge. Further, this constraint also helped in simplifying the dimensions for the regime space by providing sorting of sediments as dependent parameter on Fr_D and p . If this, what we believe to be logical, constraint was not made, an additional dimension to the regime space would be present (i.e. an axis defined by sorting). To this end, we made the following changes in the manuscript { lines 293-313 },

“For turbidity currents to propagate any significant distance, they must generate enough turbulence to enable suspension of sediments, referred to as the Knapp-Bagnold criterion^{18,53} ($U > W_{s/S}$). We minimize deposition of sediment at the inlet-fan transition by enforcing a constraint on the largest sediment size (D_{99}) at the inlet, such that $w_{s,D99} \leq u^*$. This constraint leads to emerging trends in sorting, which becomes a parameter dependent on choices of Fr_D and p at the inlet (Fig. 10, extended). We find that sorting influences the fraction of a fan surface covered by active channels and the style and rate of channel mobility. Submarine fans constructed by low Fr_D and p , coupled with poor sediment sorting, show a higher degree of channelization and development of distributary networks. The development of these complex networks can be linked to the wide range of settling velocities for the particles in transport, which leads to spatially variable deposition and erosion rates. This generates multiple nucleation sites for channel initiation and the formation of mouth bars that drive channel bifurcation (Fig. 3c). We

note that both degree of channelization and the distributary nature of the emergent channels decreases when Fr_D and p increase, as flows become well sorted. In these fans, the incoming flows rapidly become purely depositional, suppressing the growth of complex channel patterns. We note that at extremely high Fr_D values channelization is lost as most sediment bypasses the domain, hindering levee development that confines flows (Fig. 2). Future investigations could explore different criteria to define input grain size distributions, for example, keeping sorting constant across a spectrum of Fr_D and p , which may possibly quantify further the influence of sorting on emergent patterns of channel networks on the fan surface.”

REFERENCES CITED

- Dorrell, R. M., Darby, S. E., Peakall, J., Sumner, E. J., Parsons, D. R., & Wynn, R. B. (2014). The critical role of stratification in submarine channels: implications for channelization and long runout of flows. *Journal of Geophysical Research: Oceans*, 119(4), 2620-2641.
- Fildani, A., Kostic, S., Covault, J. A., Maier, K. L., Caress, D. W., & Paull, C. K. (2021). Exploring a new breadth of cyclic steps on distal submarine fans. *Sedimentology*, 68(4), 1378-1399.
- Caldwell, R. L., & Edmonds, D. A. (2014). The effects of sediment properties on deltaic processes and morphologies: A numerical modeling study. *Journal of Geophysical Research: Earth Surface*, 119(5), 961-982.
- Nienhuis, J. H., Ashton, A. D., Edmonds, D. A., Hoitink, A. J. F., Kettner, A. J., Rowland, J. C., & Törnqvist, T. E. (2020). Global-scale human impact on delta morphology has led to net land area gain. *Nature*, 577(7791), 514-518.
- Piper, D. J., & Normark, W. R. (2001). Sandy fans—from Amazon to Hueneme and beyond. *AAPG bulletin*, 85(8), 1407-1438.
- Stevenson, C. J., Feldens, P., Georgiopoulou, A., Schönke, M., Krastel, S., Piper, D. J., ... & Mosher, D. (2018). Reconstructing the sediment concentration of a giant submarine gravity flow. *Nature communications*, 9(1), 1-7.
- Parsons, J. D., Schweller, W. J., Stelting, C. W., Southard, J. B., Lyons, W. J., & Grotzinger, J. P. (2002). A preliminary experimental study of turbidite fan deposits. *Journal of Sedimentary Research*, 72(5), 619-628.
- Rowland, J., et al., 2010, A test of submarine leveed channel initiation by deposition alone. *Journal of Sedimentary Research*, 80, 710-727.
- De Leeuw, J., Eggenhuisen, J. T., & Cartigny, M. J. (2016). Morphodynamics of submarine channel inception revealed by new experimental approach. *Nature communications*, 7(1), 1-7.
- Cantelli, A., Pirmez, C., Johnson, S., & Parker, G. (2011). Morphodynamic and stratigraphic evolution of self-channelized subaqueous fans emplaced by turbidity currents. *Journal of Sedimentary Research*, 81(3), 233-247.

- Fildani, A., Normark, W. R., Kostic, S., & Parker, G. (2006). Channel formation by flow stripping: Large-scale scour features along the Monterey East Channel and their relation to sediment waves. *Sedimentology*, *53*(6), 1265-1287.
- Mutti, E., & Normark, W. R. (1987). Comparing examples of modern and ancient turbidite systems: problems and concepts. In *Marine clastic sedimentology* (pp. 1-38). Springer, Dordrecht.
- Garcia, M., & Parker, G. (1989). Experiments on hydraulic jumps in turbidity currents near a canyon-fan transition. *Science*, *245*(4916), 393-396.
- Parker, G., Fukushima, Y., & Pantin, H. M. (1986). Self-accelerating turbidity currents. *Journal of Fluid Mechanics*, *171*, 145-181.
- Pratson, L. F., & Haxby, W. F. (1996). What is the slope of the US continental slope?. *Geology*, *24*(1), 3-6.
- Azpiroz-Zabala, M., Cartigny, M. J., Talling, P. J., Parsons, D. R., Sumner, E. J., Clare, M. A., ... & Pope, E. L. (2017). Newly recognized turbidity current structure can explain prolonged flushing of submarine canyons. *Science advances*, *3*(10), e1700200.
- Babonneau, N., Savoye, B., Cremer, M., & Bez, M. (2010). Sedimentary architecture in meanders of a submarine channel: detailed study of the present Congo turbidite channel (Zaiango project). *Journal of Sedimentary Research*, *80*(10), 852-866.
- Simmons, S. M., Azpiroz-Zabala, M., Cartigny, M. J. B., Clare, M. A., Cooper, C., Parsons, D. R., ... & Talling, P. J. (2020). Novel acoustic method provides first detailed measurements of sediment concentration structure within submarine turbidity currents. *Journal of Geophysical Research: Oceans*, *125*(5), e2019JC015904.
- Paola, C., & Martin, J. M. (2012). Mass-balance effects in depositional systems. *Journal of Sedimentary Research*, *82*(6), 435-450.
- Strong, N., Sheets, B. A., Hickson, T. A., & Paola, C. (2005). A mass-balance framework for quantifying downstream changes in fluvial architecture. In *Fluvial sedimentology VII* (Vol. 35, pp. 243-253). Special Publication 35: International Association of Sedimentologists.
- Yu, L., Li, Q., & Straub, K. M. (2017). Scaling the response of deltas to relative-sea-level cycles by autogenic space and time scales: a laboratory study. *Journal of Sedimentary Research*, *87*(8), 817-837.

Reviewer #1 (Remarks to the Author):

Dear Authors and Editor,

I have re-reviewed the manuscript "A dimensionless framework for predicting submarine fan morphology." I think the Authors did a decent job at addressing some of the reviewers' concerns.

There are still few points that should be clarified or at least made.

I am a bit worried when I read certain things in the rebuttal (and still lingering in the manuscript). Deep-sea fans exploration is still on an early phase and fans' mapping is still quite limited and done at very coarse scale (which gives us pretty coarse measurements of slopes). There is a risk at binning fans as subcritical and supercritical for many reasons, one being that this could be proven a false dichotomy relatively soon. I say so because we have been learning so much in the last decade that any of the previous classifications became obsolete. Saying that a fan like the Bengal, of an extent circa 3.0×10^6 km² and very much mapped at a coarse scale, is 'all' subcritical is extremely perilous mostly because it has sectors that might be traversed by supercritical flows. Binning large swaths of the seafloor with sweeping definitions is a bit inattentive (at least for a marine geologist that senses the much work ahead.) Cyclic steps are spelled correctly in the text. Thank you! Still missing the citation Fildani et al. 2006 even though it is mentioned as added in the rebuttal. I do think the 2006 manuscript is important and should be cited because cyclic steps (recognized in 2006 as net-erosional and net depositional) do tend to migrate updip and might dictate avulsions.

I have noticed (in the Methods) that the Authors use a personal communication from John Hughes Clarke. It is fine by me but it might become an issue later in production. My Regards to the Authors.

Reviewer #2 (Remarks to the Author):

I would like to commend the authors for the additional, and thorough, work made in responding to reviewers' comments. I believe the manuscript makes a substantial advance of broad interest, providing a parameter space analysis of characteristics of material transport in deep marine sedimentary systems.

I recommend the eventual publication of this work in Nature Communications. However, following the reviewers' comments [see below] there are a few key points that need to be addressed.

**Robert Dorrell
28/05/2022**

Comments

L272 I am still concerned by the excessive deposition seen in the simulation results and the implications on flow dynamics. It would be very informative to have plots in the manuscript [or supplementary] that show relative bed depth [ie inclusive of initial slope] and plots that show streamwise slope. Could the authors test if initial deposit build up $O(100m)+$ blocks flow and/or results in in a hydraulic jump (or otherwise exacerbates flow deceleration) enhancing proximal sediment deposition. If so what do "numerical tank" experiments need to do to address this [potential] issue.

L275 The introduction of analysis of intermittent flows is excellent, as are the supplementary videos (see note below). However, I believe there is a more fundamental change in observed dynamics and morphology than that just captured by change in rugosity. Figure 9 suggests that intermittent flows consistently build "stronger" channels (where the thalweg to levee crest depth appears bigger).

Can you test if this is correct, quantify the change including the impact on downstream flux and discuss the potential implications.

This (the role of intermittent flows, with shear in the head) in simulations may be linked to work on channel flushing (Heijnen et al. 2022) emptying loose sediment fill blocking a channel. Is there a difference in the number, frequency or magnitude of channel avulsions between continuous and intermittent flow?

L295 if the form of KB criterion $U > w_s / S$ is assumed it must be noted that this is in the limit of negligible entrainment:

Energy production exceeds work done keeping sediment in suspension
 $u^*{}^2 U > g R c h w_s$

skin + [entrainment] drag balances gravitational acceleration [Abad et al, 2011]
 $u^*{}^2 + e U^2 (1+Ri/2) \sim g R c h S$

Supplementary Material It is noted that in the supplementary video the colormap range for the left-hand bathymetric plot varies on a frame by frame basis. Please set this to a consistent range.

References

Abad, J. D., Sequeiros, O. E., Spinewine, B., Pirmez, C., Garcia, M. H., & Parker, G. (2011). Secondary current of saline underflow in a highly meandering channel: experiments and theory. *Journal of Sedimentary Research*, 81(11), 787-813.

Heijnen, M. S., Clare, M. A., Cartigny, M. J., Talling, P. J., Hage, S., Pope, E. L., ... & Clarke, J. E. H. (2022). Fill, flush or shuffle: How is sediment carried through submarine channels to build lobes?. *Earth and Planetary Science Letters*, 584, 117481.

Reviewer #1 (Remarks to the Author):

Dear Authors and Editor,

I have re-reviewed the manuscript “A dimensionless framework for predicting submarine fan morphology.” I think the Authors did a decent job at addressing some of the reviewers’ concerns. There are still few points that should be clarified or at least made.

AW: We thank reviewer #1 for their second round of reviews on our manuscript. We appreciate the reviewer’s applause for addressing the reviewers concerns in the first round of reviews. We hope that with this second attempt we address their remaining concerns.

I am a bit worried when I read certain things in the rebuttal (and still lingering in the manuscript). Deep-sea fans exploration is still on an early phase and fans’ mapping is still quite limited and done at very coarse scale (which gives us pretty coarse measurements of slopes). There is a risk at binning fans as subcritical and supercritical for many reasons, one being that this could be proven a false dichotomy relatively soon. I say so because we have been learning so much in the last decade that any of the previous classifications became obsolete. Saying that a fan like the Bengal, of an extent circa 3.0×10^6 km² and very much mapped at a coarse scale, is ‘all’ subcritical is extremely perilous mostly because it has sectors that might be traversed by supercritical flows. Binning large swaths of the seafloor with sweeping definitions is a bit inattentive (at least for a marine geologist that senses the much work ahead.)

AW: We thank the reviewer for raising this valid concern. While it is true that current seafloor mapping provides coarse measurements of local slopes on a submarine fan, they still capture accurate regional bathymetric gradients {Pratson and Haxby 1996}, which is what we compute from a choice of Fr_D and set as the initial bathymetric surface in our workflow. In our manuscript we do not claim anywhere that a subcritical fan maintains a subcritical Fr_D state everywhere. In fact, in our manuscript we make a point that in our subcritical fan simulations flow over levees change criticality and induce the formation of cyclic steps {see line 147-152}. Cyclic steps are formed during hydraulic jump conditions where the Fr_D fluctuates between greater than 1 to less than 1. This transition in the Fr_D state can be clearly observed in the subcritical simulation movie as well. We concur with the statement that the Bengal fan might have supercritical features along some segments of the fan surface. We also report supercritical bedforms such as cyclic steps on the levees of our subcritical fan simulation. To clarify further, we do not propose that a subcritical fan must maintain subcritical flow everywhere, more so, we propose that the overall fan morphology is constructed by flows within submarine channels that maintains a mean Fr_D state that is subcritical. We now also provide 1D plots of several fans that make this point, where flow within channels of a subcritical fan maintains a mean subcritical Fr_D state {Fig. 12-13, extended}. This lends support to our argument that Fr_D and p conditions in canyons or relatively immobile channels imparts the first order control on the morphology of submarine fans and the properties of the flows in their emergent channels. To make this point more succinct in the manuscript we now add the following text {see line 147-152}.

“While we force the simulations with inlet subcritical ($Fr_D < 1$) flow, the flow behavior over the emergent fan is not subcritical everywhere on the fan surface. For example, on the

levees, flow tends to locally change criticality and induce formation of cyclic steps⁴⁷ (Fig. 3d & subcritical simulation movie). However, the average flow conditions in the inlet channel and the emergent self-formed channels maintain a mean subcritical ($Fr_D < 1$) state.”

Cyclic steps are spelled correctly in the text. Thank you! Still missing the citation Fildani et al. 2006 even though it is mentioned as added in the rebuttal. I do think the 2006 manuscript is important and should be cited because cyclic steps (recognized in 2006 as net-erosional and net depositional) do tend to migrate up dip and might dictate avulsions.

AW: We thank the reviewer for this comment. We initially cited a recent publication (Fildnai et al 2022) in reference to cyclic steps. We now cite Fildani et al 2006 {see line 150}.

I have noticed (in the Methods) that the Authors use a personal communication from John Hughes Clarke. It is fine by me but it might become an issue later in production. My Regards to the Authors.

AW: We thank the reviewer for this comment. We assume the editor will notify us if this is a concern for Nature publications. If so, we hope the editor can guide us on how to navigate this matter given their prior experience.

Reviewer #2 (Remarks to the Author):

I would like to commend the authors for the additional, and thorough, work made in responding to reviewers’ comments. I believe the manuscript makes a substantial advance of broad interest, providing a parameter space analysis of characteristics of material transport in deep marine sedimentary systems.

I recommend the eventual publication of this work in Nature Communications. However, following the reviewers’ comments [see below] there are a few key points that need to be addressed.

Robert Dorrell
28/05/2022

AW: We thank the reviewer for their second constructive review. We found his reviews very insightful, which have greatly strengthened the work in this manuscript.

Comments

L272 I am still concerned by the excessive deposition seen in the simulation results and the implications on flow dynamics. It would be very informative to have plots in the manuscript [or supplementary] that show relative bed depth [ie inclusive of initial slope] and plots that show streamwise slope. Could the authors test if initial deposit build up O(100m)+ blocks flow and/or results in a hydraulic jump (or otherwise exacerbates flow deceleration) enhancing proximal sediment deposition. If so what do “numerical tank” experiments need to do to address this [potential] issue.

AW: We thank the reviewer for this excellent comment. We would like to highlight that not all simulated fans show high initial build ups {see Fig. 2}. Some models completely bypass/erode

any sediments within the inlet channel. A smaller subset of the fans do have high initial build up within and downstream of the inlet channel, particularly ones that are characterized by a high p and low Fr_D (relatively depositional flows). In general, we see a gradient in our regime space for the degree of input sediment that bypasses the domain. This degree of bypass is likely due to our method for defining the initial slope, which is a function of input Fr_D but not input p . As mentioned in the previous review round, we are unaware of any formulation for regional slope that takes into account both the Fr_D and p conditions. This highlights a need for our community to address, but here we utilize a standard and well cited approach.

To further address the reviewer's comment, we analyzed several fans from the regime diagram and show 1D plots that displays relative bed depths and Fr_D and p state within the inlet channel, all the way to its loss of confinement{Fig. 12-16, extended}. These plots show that the net bathymetry within the inlet channel does not block flows or exacerbate flow deceleration enhancing proximal sedimentation. Nor do they form hydraulic jumps that maintain a set location in the simulations.

L275 The introduction of analysis of intermittent flows is excellent, as are the supplementary videos (see note below). However, I believe there is a more fundamental change in observed dynamics and morphology than that just captured by change in rugosity. Figure 9 suggests that intermittent flows consistently build “stronger” channels (where the thalweg to levee crest depth appears bigger). Can you test if this is correct, quantify the change including the impact on downstream flux and discuss the potential implications. This (the role of intermittent flows, with shear in the head) in simulations may be linked to work on channel flushing (Heijnen et al. 2022) emptying loose sediment fill blocking a channel. Is there a difference in the number, frequency or magnitude of channel avulsions between continuous and intermittent flow?

AW: We appreciate the reviewer for applauding our added analyses on flow intermittency and the ensuing excellent comment about its impact on submarine fan channel dynamics. Here we highlight again that changing flow intermittency did not result in significant changes in fan rugosity, which quantifies the planform complexity of fans and is the main focus in this manuscript. We do note that intermittent flows as compared to steady flows built “stronger” or at least deeper channels. Intermittent flows introduces more frequent current heads that are characterized by high shear velocities, which help sediments to remain in flux. These observed dynamics could influence the time necessary for a submarine channel to avulse and may have potential implication for the stratigraphic architecture. While this might sound odd, we are not aware to date of a quantitative definition of an avulsion event that can help us quantify the impact of flow intermittency on submarine fan channel dynamics. By this, we mean a method that can be used to isolate avulsions and separate them from slow lateral channel migration. Many avulsions play out over some defined timescale, that can make them hard to isolate with quantitative identification methods used on the type of maps and flow data we have here. Further work is needed to quantitatively define an avulsion event before exploring the role of flow intermittency on submarine channel dynamics. Therefore, we find this interesting question to be outside of the scope we attempt to set in this manuscript. Perhaps a subsequent article fully dedicated to exploring the influence of flow intermittency on submarine channel dynamics and

their resultant stratigraphy would be a more suitable and a just approach. However, to acknowledge this excellent comment, we now add the following text {line 281-289} in the manuscript, briefly highlighting the influence of flow intermittency on submarine fan channel dynamics.

“We find that flow intermittency does not significantly influence fan rugosity, with values changing due to flow intermittency by 2-10%. However, we note that intermittent flows built relatively deeper channels. Intermittent flows introduce relatively more frequent current heads that possess relatively higher shear velocities, which help maintain sediment in suspension. This phenomenon suppresses sedimentation rates within channels and could potentially impact the frequency or magnitude of avulsions on submarine fans and the resulting stratigraphic architecture. We show that flow intermittency appears to be a second order control on fan shape, when compared to the influence of Fr_D and p (Fig. 9 a-b).”

L295 if the form of KB criterion $U > ws / S$ is assumed it must be noted that this is in the limit of negligible entrainment:

Energy production exceeds work done keeping sediment in suspension
 $u^*^2 U > g R c h ws$

skin + [entrainment] drag balances gravitational acceleration [Abad et al, 2011]
 $u^*^2 + e U^2 (1+Ri/2) \sim g R c h S$

AW: We thank the reviewer for this comment. We agree that KB criterion is in the limit of negligible entrainment and we acknowledge more recent and rigorous criteria, such as the one highlighted here (Abad et al, 2011). We make a correction by removing the explicit reference to KB criterion, which might mislead the reader in assuming that it's the benchmark of sediment suspension via turbulence within turbidity currents {see line 302-303}.

“For turbidity currents to propagate any significant distance, they must generate enough turbulence to enable suspension of sediments^{15,55}.”

Supplementary Material It is noted that in the supplementary video the colormap range for the left-hand bathymetric plot varies on a frame by frame basis. Please set this to a consistent range.

AW: We thank the reviewer for this comment and we now use a fix range on the color map for the bathymetric maps.

References

Pratson, L. F., & Haxby, W. F. (1996). What is the slope of the US continental slope?. *Geology*, 24(1), 3-6.

Reviewer #1 (Remarks to the Author):

I think the Authors responded to most of my concerns. I do not see many reasons to hold this manuscript back. I support its publication.

I do have a minor concern about their seafloor descriptions: the Authors describe biforcations on modern fans (using low quality data), are we sure these are biforcations? Could they be avulsion nodes? (as shown by higher resolution datasets ignored by the Authors). By watching carefully their videos, I think these might be avulsion nodes (hence there are no mouthbars forming... and so on).

Other than that... Best of luck to the Authors.

Reviewer #2 (Remarks to the Author):

I would like to thank the authors for their thorough review and response to comments. The work detailed and is of excellent standard. The additional supplementary material provides both context and confidence in the methodology and model results.

The work outlining the role of intermittency on structural morphodynamics could have had further quantitative analysis. Although the authors quantify it as 2nd order importance (in terms of rugosity), longer time studies of evolution could have been useful - given propensity to form deeper channels. However, as the authors detail in their response to review this is a new field of research beyond the current manuscript. Indeed, this could have been detailed in the manuscript itself.

I fully support the publication of this manuscript
Robert Dorrell
01/08/2022

Round – 3: REVIEWERS' COMMENTS

Reviewer #1 (Remarks to the Author):

I think the Authors responded to most of my concerns. I do not see many reasons to hold this manuscript back. I support its publication.

I do have a minor concern about their seafloor descriptions: the Authors describe bifurcations on modern fans (using low quality data), are we sure these are bifurcations? Could they be avulsion nodes? (as shown by higher resolution datasets ignored by the Authors). By watching carefully their videos, I think these might be avulsion nodes (hence there are no mouth bars forming... and so on).

Other than that... Best of luck to the Authors.

Reviewer #2 (Remarks to the Author):

I would like to thank the authors for their thorough review and response to comments. The work detailed and is of excellent standard. The additional supplementary material provides both context and confidence in the methodology and model results.

The work outlining the role of intermittency on structural morphodynamics could have had further quantitative analysis. Although the authors quantify it as 2nd order importance (in terms of rugosity), longer time studies of evolution could have been useful - given propensity to form deeper channels. However, as the authors detail in their response to review this is a new field of research beyond the current manuscript. Indeed, this could have been detailed in the manuscript itself.

I fully support the publication of this manuscript

Robert Dorrell

01/08/2022

Round - 3: Response to reviewers comments on ‘A dimensionless framework for predicting submarine fan morphology’

Reviewer #1 (Remarks to the Author):

I think the Authors responded to most of my concerns. I do not see many reasons to hold this manuscript back. I support its publication.

AW: On behalf of all the authors, we would like to thank you for taking the time to reviewing our manuscript and presenting your feedback.

I do have a minor concern about their seafloor descriptions: the Authors describe bifurcations on modern fans (using low quality data), are we sure these are bifurcations? Could they be avulsion nodes? (as shown by higher resolution datasets ignored by the Authors). By watching carefully their videos, I think these might be avulsion nodes (hence there are no mouth bars forming... and so on).

AW: We thank the reviewer for this comment. This comment on bifurcation is related to subcritical fans. As we have highlighted in the first round of revision, we focus on comparing endmember modeled fans to examples of modern fans with equivalent gradients and forcing conditions to show the efficacy of our modeling approach in generating realizations of remarkably similar morphologies. To date, we do not have bathymetric maps in low gradient systems (i.e. Zaire, Congo, Indus, and Amazon) of the resolution found on high gradient settings (i.e. La Jolla fan). It is not possible to delineate the observed downstream branching on the Modern Zaire fan surface whether they are bifurcations or avulsions nodes without extensive field velocity measurements. This is where the model can help us interpret these preserved features. In the subcritical simulation movie, the velocity field clearly shows that these branches are bifurcations as both channels downstream of branching node contain active flow {see subcritical simulation movie at time 00:06;00:14;00:35;01:07 and so on}

Other than that... Best of luck to the Authors.

AW: Thank you.

Reviewer #2 (Remarks to the Author):

I would like to thank the authors for their thorough review and response to comments. The work detailed and is of excellent standard. The additional supplementary material provides both context and confidence in the methodology and model results.

AW: On behalf of all the authors on this manuscript, we would like thank you for taking the time and carefully reviewing our manuscript. We believe that your reviews/feedback have greatly helped us in strengthening this work.

The work outlining the role of intermittency on structural morphodynamics could have had further quantitative analysis. Although the authors quantify it as 2nd order importance (in terms of rugosity), longer time studies of evolution could have been useful - given propensity to form deeper channels. However, as the authors detail in their response to review this is a new field of research beyond the current manuscript. Indeed, this could have been detailed in the manuscript itself. I fully support the publication of this manuscript.

AW: We thank the reviewer for this comment. We agree that the role of intermittent flow could have potential impacts on the stratigraphic architecture of submarine fans. However, in this study, as we focused on exploring forcing conditions that sets the shape of submarine fans, we found that the role of intermittent flow only imparted a second order control. Indeed, this is an exciting new question that requires definition and development of new metrics and methods to properly quantify the role of intermittent flows on submarine fan architecture. The authors are excited to tackle this very question in standalone manuscript.

AW: We are grateful to the reviewers for helping us improve this manuscript through several rounds of revisions. Finally, we would also like to thank all the editorial staff involved in

facilitating the peer review process of this manuscript. We greatly enjoyed this process and are looking forward to this article getting published in Nature Communications.